# On the Trade-Off between Actionable Explanations and the Right to be Forgotten

**Martin Pawelczyk**[1]*, **Tobias Leemann**[1], **Asia Biega**[2]†, **and Gjergji Kasneci**[1]†
[1]University of Tübingen, Germany
[2]Max-Planck-Institute for Security and Privacy, Germany

## Abstract

As machine learning (ML) models are increasingly being deployed in high-stakes applications, policymakers have suggested tighter data protection regulations (e.g., GDPR, CCPA). One key principle is the "right to be forgotten" which gives users the right to have their data deleted. Another key principle is the right to an actionable explanation, also known as algorithmic recourse, allowing users to reverse unfavorable decisions. To date, it is unknown whether these two principles can be operationalized simultaneously. Therefore, we introduce and study the problem of recourse invalidation in the context of data deletion requests. More specifically, we theoretically and empirically analyze the behavior of popular state-of-the-art algorithms and demonstrate that the recourses generated by these algorithms are likely to be invalidated if a small number of data deletion requests (e.g., 1 or 2) warrant updates of the predictive model. For the setting of differentiable models, we suggest a framework to identify a minimal subset of critical training points which, when removed, maximize the fraction of invalidated recourses. Using our framework, we empirically show that the removal of as little as 2 data instances from the training set can invalidate up to 95 percent of all recourses output by popular state-of-the-art algorithms. Thus, our work raises fundamental questions about the compatibility of "the right to an actionable explanation" in the context of the "right to be forgotten", while also providing constructive insights on the determining factors of recourse robustness.

## 1 Introduction

Machine learning (ML) models make a variety of consequential decisions in domains such as finance, healthcare, and policy. To protect users, laws such as the European Union's General Data Protection Regulation (GDPR) (GDPR, 2016) or the California Consumer Privacy Act (CCPA) (OAG, 2021) constrain the usage of personal data and ML model deployments. For example, individuals who have been adversely impacted by the predictions of these models have the right to *recourse* (Voigt & Von dem Bussche, 2017), i.e., a constructive instruction on how to act to arrive at a more desirable outcome (e.g., change a model prediction from "loan denied" to "approved"). Several approaches in recent literature tackled the problem of providing recourses by generating instance level counterfactual explanations (Wachter et al., 2018; Ustun et al., 2019; Karimi et al., 2020; Pawelczyk et al., 2020a).

Complementarily, data protection laws provide users with greater authority over their personal data. For instance, users are granted the right to *withdraw consent to the usage of their data* at any time (Biega & Finck, 2021). These regulations affect technology platforms that train their ML models on personal user data under the respective legal regime. Law scholars have argued that the continued use of ML models relying on deleted data instances could be deemed illegal (Villaronga et al., 2018).

Irrespective of the underlying mandate, data deletion has raised a number of algorithmic research questions. In particular, recent literature has focused on the efficiency of deletion (i.e., how to delete individual data points without retraining the model (Ginart et al., 2019; Golatkar et al., 2020a)) and model accuracy aspects of data deletion (i.e., how to remove data without compromising model

---

*Corresponding author: martin.pawelczyk@uni-tuebingen.de
†Equal senior author contribution.

accuracy (Biega et al., 2020; Goldsteen et al., 2021)). An aspect of data deletion which has not been examined before is *whether and how data deletion may impact model explanation frameworks*. Thus, there is a need to understand and systematically characterize the limitations of recourse algorithms when personal user data may need to be deleted from trained ML models. Indeed, deletion of certain data instances might invalidate actionable model explanations – both for the deleting user and, critically, unsuspecting other users. Such invalidations can be especially problematic in cases where users have already started to take costly actions to change their model outcomes based on previously received explanations.

In this paper, we formally examine the problem of algorithmic recourse in the context of data deletion requests. We consider the setting where a small set of individuals has decided to withdraw their data and, as a consequence of the deletion request, the model needs to be updated (Ginart et al., 2019). In particular, this work tackles the subsequent pressing question:

> *What is the worst impact that a deleted data instance can have on the recourse validity?*

We approach this question by considering two distinct scenarios. The first setting considers to what extent the outdated recourses still lead to a desirable prediction (e.g., loan approval) on the updated model. For this scenario, we suggest a robustness measure called *recourse outcome instability* to quantify the fragility of recourse methods. Second, we consider the setting where the recourse action is being updated as a consequence of the prediction model update. In this case, we study what maximal change in recourse will be required to maintain the desirable prediction. To quantify the extent of this second problem, we suggest the notion of *recourse action instability*.

Given these robustness measures, we derive and analyze theoretical worst-case guarantees of the maximal instability induced for linear models and neural networks in the overparameterized regime, which we study through the lens of neural tangent kernels. We furthermore define an optimization problem for empirically quantifying recourse instability under data deletion. For a given trained ML model, we identify small sets of data points that maximize the proposed instability measures when deleted. Since the resulting brute-force approach (i.e., retraining models for every possible removal set) is NP-hard, we propose two relaxations for recourse instability maximization that can be optimized using (i) end-to-end gradient descent or (ii) via a greedy approximation algorithm. To summarize, in this work we make the following key contributions:

- **Novel recourse robustness problem.** We introduce the problem of *recourse invalidation under the right to be forgotten* by defining two new recourse instability measures.
- **Theoretical analysis.** Through rigorous theoretical analysis, we identify the factors that determine the instability of recourses when users whose data is part of the training set submit deletion requests.
- **Tractable algorithms.** Using our instability measures, we present an optimization framework to identify a small set of critical training data points which, when removed, invalidates most of the issued recourses.
- **Comprehensive experiments.** We conduct extensive experiments on multiple real-world data sets for both regression and classification tasks with our proposed algorithms, showing that the removal of even one point from the training set can invalidate up to 95 percent of all recourses output by state-of-the-art methods

Our results also have practical implications for system designers. First, our analysis and algorithms help identify parameters and model classes leading to higher stability when a trained ML model is subjected to deletion requests. Furthermore, our proposed methods can provide an informed way towards practical implementations of data minimization (Finck & Biega, 2021), as one could argue that data points contributing to recourse instability could be minimized out. Hence, our methods could increase designer's awareness and the compliance of their trained models.

## 2 RELATED WORK

**Algorithmic Approaches to Recourse.** Several approaches in recent literature have been suggested to generate recourse for users who have been negatively impacted by model predictions (Tolomei et al., 2017; Laugel et al., 2017; Dhurandhar et al., 2018; Wachter et al., 2018; Ustun et al., 2019;

Van Looveren & Klaise, 2019; Pawelczyk et al., 2020a; Mahajan et al., 2019; Mothilal et al., 2020; Karimi et al., 2020; Rawal & Lakkaraju, 2020; Dandl et al., 2020; Antorán et al., 2021; Spooner et al., 2021; Albini et al., 2022). These approaches generate recourses assuming a static environment without data deletion requests, where both the model and the recourse remain stable.

A related line of work has focused on determining the extent to which recourses remain invariant to the model choice (Pawelczyk et al., 2020b; Black et al., 2021), to data distribution shifts (Rawal et al., 2021; Upadhyay et al., 2021), perturbations to the input instances (Artelt et al., 2021; Dominguez-Olmedo et al., 2022; Slack et al., 2021), or perturbations to the recourses (Pawelczyk et al., 2023).

**Sample Deletion in Predictive Models.** Since according to EU's GDPR individuals can request to have their data deleted, several approaches in recent literature have been focusing on updating a machine learning model without the need of retraining the entire model from scratch (Wu et al., 2020; Ginart et al., 2019; Izzo et al., 2021; Golatkar et al., 2020a;b; Cawley & Talbot, 2004). A related line of work considers the problem of data valuation (Ghorbani et al., 2020; Ghorbani & Zou, 2019). Finally, removing subsets of training data is an ingredient used for model debugging (Doshi-Velez & Kim, 2017) or the evaluation of explanation techniques (Hooker et al., 2019; Rong et al., 2022).

**Contribution.** While we do not suggest a new recourse algorithm, our work addresses the problem of recourse fragility in the presence of data deletion requests, which has previously not been studied. To expose this fragility, we suggest effective algorithms to delete a minimal subset of critical training points so that the fraction of invalidated recourses due to a required model update is maximized. Moreover, while prior research in the data deletion literature has primarily focused on effective data removal strategies for predictive models, there is no prior work that studies to what extent recourses output by state-of-the-art methods are affected by data deletion requests. Our work is the first to tackle these important problems and thereby paves the way for recourse providers to evaluate and rethink their recourse strategies in light of the right to be forgotten.

## 3 PRELIMINARIES

**The Predictive Model and the Data Deletion Mechanism.** We consider prediction problems from some input space $\mathbb{R}^d$ to an output space $\mathcal{Y}$, where $d$ is the number of input dimensions. We denote a sample by $\mathbf{z} = (\mathbf{x}, y)$, and denote the training data set by $\mathcal{D} = \{\mathbf{z}_1, \ldots, \mathbf{z}_n\}$. Consider the weighted empirical risk minimization problem (ERM), which gives rise to the optimal model parameters:

$$\mathbf{w}_{\boldsymbol{\omega}} = \arg\min_{\mathbf{w}'} \sum_{i=1}^{n} \omega_i \cdot \ell\big(y_i, f_{\mathbf{w}'}(\mathbf{x}_i)\big), \tag{1}$$

where $\ell(\cdot, \cdot)$ is an instance-wise loss function (e.g., binary cross-entropy, mean-squared-error (MSE) loss, etc.) and $\boldsymbol{\omega} \in \{0, 1\}^n$ are data weights that *are fixed at training time*. If $\omega_i = 1$, then the point $\mathbf{z}_i = (\mathbf{x}_i, y_i)$ is part of the training data set, otherwise it is not. During model training, we set $\omega_i = 1 \ \forall i$, that is, the decision maker uses all available training instances at training time. In the optimization expressed in equation 1, the model parameters $\mathbf{w}$ are usually an implicit function of the data weight vector $\boldsymbol{\omega}$ and we write $\mathbf{w}_{\boldsymbol{\omega}}$ to highlight this fact; in particular, when all training instances are used we write $\mathbf{w}_{\mathbf{1}}$, where $\mathbf{1} \in \mathbb{R}^n$ is a vector of 1s. In summary, we have introduced the *weighted* ERM problem since it allows us to understand the impact of arbitrary data deletion patterns on actionable explanations as we allow users to withdraw their entire input $\mathbf{z}_i = (y_i, \mathbf{x}_i)$ from the training set used to train the model $f_{\mathbf{w}_{\mathbf{1}}}$. Next, we present the recourse model we consider.

**The Recourse Problem in the Context of the Data Deletion Mechanism.** We follow an established definition of counterfactual explanations originally proposed by Wachter et al. (2018). For a given model $f_{\mathbf{w}_{\boldsymbol{\omega}}} : \mathbb{R}^d \to \mathbb{R}$ parameterized by $\mathbf{w}$ and a distance function $d(\cdot, \cdot) : \mathcal{X} \times \mathcal{X} \to \mathbb{R}_+$, the problem of finding a recourse $\check{\mathbf{x}} = \mathbf{x} + \boldsymbol{\delta}$ for a factual instance $\mathbf{x}$ is given by:

$$\boldsymbol{\delta}_{\boldsymbol{\omega}, \mathbf{x}} \in \arg\min_{\boldsymbol{\delta}' \in \mathcal{A}_d} \left(f_{\mathbf{w}_{\boldsymbol{\omega}}}(\mathbf{x} + \boldsymbol{\delta}') - s\right)^2 + \lambda \cdot d(\mathbf{x}, \mathbf{x} + \boldsymbol{\delta}'), \tag{2}$$

where $\lambda \geq 0$ is a scalar tradeoff parameter and $s$ denotes the target score. In the optimization from equation 2, the optimal recourse action $\boldsymbol{\delta}$ usually depends on the model parameters and since the model parameters themselves depend on the exact data weights configuration we write $\boldsymbol{\delta}_{\boldsymbol{\omega}, \mathbf{x}}$ to highlight this fact. The first term in the objective on the right-hand-side of equation 2 encourages the outcome $f_{\mathbf{w}_{\boldsymbol{\omega}}}(\check{\mathbf{x}})$ to become close to the user-defined target score $s$, while the second term encourages

the distance between the factual instance $\mathbf{x}$ and the recourse $\check{\mathbf{x}}_{\boldsymbol{\omega}} := \mathbf{x} + \boldsymbol{\delta}_{\boldsymbol{\omega},\mathbf{x}}$ to be low. The set of constraints $\mathcal{A}_d$ ensures that only admissible changes are made to the factual $\mathbf{x}$.

**Recourse Robustness Through the Lens of the Right to be Forgotten.** We first introduce several key terms, namely, *prescribed recourses* and *recourse outcomes*. A prescribed recourse $\check{\mathbf{x}}$ refers to a recourse that was provided to an end user by a recourse method (e.g., salary was increased by $500). The recourse outcome $f(\check{\mathbf{x}})$ is the model's prediction evaluated at the recourse. With these concepts in place, we develop two recourse instability definitions.

**Definition 1.** *(Recourse outcome instability) The recourse outcome instability with respect to a factual instance* $\mathbf{x}$*, where at least one data weight is set to* $0$*, is defined as follows:*

$$\Delta_{\mathbf{x}}(\boldsymbol{\omega}) = \left| f_{\mathbf{w_1}}(\check{\mathbf{x}}_1) - f_{\mathbf{w_{\omega}}}(\check{\mathbf{x}}_1) \right|, \tag{3}$$

*where* $f_{\mathbf{w_1}}(\check{\mathbf{x}}_1)$ *is the prediction at the prescribed recourse* $\check{\mathbf{x}}_1$ *based on the model that uses the full training set (i.e.,* $f_{\mathbf{w_1}}$*) and* $f_{\mathbf{w_{\omega}}}(\check{\mathbf{x}}_1)$ *is the prediction at the prescribed recourse for an updated model and data deletion requests have been incorporated into the predictive model (i.e.,* $f_{\mathbf{w_{\omega}}}$*).*

The above definition concisely describes the effect of applying "outdated" recourses to the updated model. We assume that only the model parameters are being updated while the prescribed recourses remain unchanged. For a discrete model with $\mathcal{Y} = \{0, 1\}$, Definition 1 captures whether the prescribed recourses will be invalid ($\Delta_{\mathbf{x}} = 1$) after deletion of training instances (see Fig. 1a). To obtain invalidation rates of recourses for a continuous-score model with target value $s$, we can also apply Definition 1 with a discretized $f'(\mathbf{x}) = \mathbb{I}\left[ f(\mathbf{x}) > s \right]$, where $\mathbb{I}$ denotes the indicator function.

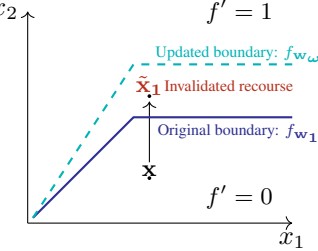

(a) Recourse Outcome Instability

In Definition 2, consistent with related work (e.g., Wachter et al. (2018)), the distance function $d$ is specified to be a p-norm and the recourse is allowed to change due to model parameter updates.

**Definition 2.** *(Recourse action instability) The Recourse action instability with respect to a factual input* $\mathbf{x}$*, where at least one data weight is set to* $0$*, is defined as follows:*

$$\Phi_{\mathbf{x}}^{(p)}(\boldsymbol{\omega}) = \left\| \check{\mathbf{x}}_1 - \check{\mathbf{x}}_{\boldsymbol{\omega}} \right\|_p, \tag{4}$$

*where* $p \in [1, \infty)$*, and* $\check{\mathbf{x}}_{\boldsymbol{\omega}}$ *is the recourse obtained for the model trained on the data instances that remain present in the data set after the deletion request.*

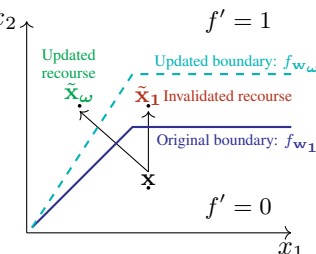

(b) Recourse Action Instability

Definition 2 quantifies the extent to which the prescribed recourses would have to additionally change to still achieve the desired recourse outcome *after data deletion* requests (i.e., $\check{\mathbf{x}}_{\boldsymbol{\omega}}$, see Fig. 1b). Note that we are interested in how the optimal low cost recourse changes even if the outdated recourse would remain valid. Using our invalidation measures defined above, in the next section, we formally study the trade-offs between actionable explanations and the right to be forgotten. To do so, we provide data dependent upper bounds on the invalidation measures from Definitions 1 and 2, which practitioners can use to probe the worst-case vulnerability of their algorithmic recourse to data deletion requests.

Figure 1: Visualizing the two key robustness notions. In Fig. 1a, recourse $\check{\mathbf{x}}_1$ for an input $\mathbf{x}$ is invalidated due to a model update. In Fig. 1b, recourse is additionally recomputed (i.e., $\check{\mathbf{x}}_{\boldsymbol{\omega}}$) to avoid recourse invalidation.

## 4  TRADE-OFFS BETW. ALG. RECOURSE AND THE RIGHT TO BE FORGOTTEN

Here we relate the two notions of recourse instability presented in Definitions 1 and 2 to the vulnerability of the underlying predictive model with respect to data deletion requests. We show that the introduced instability measures are directly related to data points with a high influence on the parameters after deletion.

**Analyzing the Instability Measures.** With the basic terminology in place, we provide upper bounds for the recourse instability notions defined in the previous section when the underlying models are

linear or overparameterized neural networks. Throughout our analysis, we the term $\mathbf{d}_i := \mathbf{w_1} - \mathbf{w}_{-i}$ has an important impact. It measures the difference from the original parameters $\mathbf{w_1}$ to the parameter vector $\mathbf{w}_{-i}$ obtained after deleting the $i$-th instance $(\mathbf{x}_i, y_i)$ from the model. In the statistics literature, this term is also known as the empirical influence function (Cook & Weisberg, 1980). Below we provide an upper bound for recourse outcome instability in linear models.

**Proposition 1** (Upper bound on recourse outcome instability for linear models). *For the linear regression model* $f(\mathbf{x}) = \mathbf{w}_L^\top \mathbf{x}$ *with model parameters* $\mathbf{w}_L = (\mathbf{X}^\top \mathbf{X})^{-1}\mathbf{X}^\top \mathbf{Y}$, *an upper bound for the recourse invalidation from Definition 1 by removing an instance from the training set is given by:*

$$\Delta_{\mathbf{x}} \leq \|\check{\mathbf{x}}_{\mathbf{1}}\|_2 \cdot \max_{i \in [n]} \|\mathbf{d}_i^L\|_2, \tag{5}$$

*where* $\mathbf{d}_i^L := \mathbf{w}_L - \mathbf{w}_{L,-i} = (\mathbf{X}^\top \mathbf{X})^{-1}\mathbf{x}_i \cdot \frac{r_i}{1 - h_{ii}}$, $r_i = y_i - \mathbf{w}_L^\top \mathbf{x}_i$ *and* $h_{ii} = \mathbf{x}_i^\top (\mathbf{X}^\top \mathbf{X})^{-1}\mathbf{x}_i$.

The term $(\mathbf{X}^\top \mathbf{X})^{-1}\mathbf{x}_i = \frac{d\mathbf{w}_L}{dy_i}$ describes how sensitive the model parameters are to $y_i$, while the residual $r_i$ captures how well $y_i$ can be fit by the model. On the contrary, the term $h_{ii}$ from the denominator is known as the *leverage* and describes how atypical $\mathbf{x}_i$ is with respect to all training inputs $\mathbf{X}$. In summary, data instances that have influential labels or are atypical will have the highest impact when deleted. Next, we provide a generic upper bound on recourse action instability.

**Proposition 2** (Upper bound on recourse action instability). *For any predictive model with scoring function* $f : \mathbb{R}^d \to \mathbb{R}$, *an upper bound for the recourse instability from Definition 2 by removing an instance* $\mathbf{z}_i = (\mathbf{x}, y)$ *from the training set is given by:*

$$\Phi_{\mathbf{x}}^{(2)} \leq \|\mathbf{d}_i\|_2 \int_0^1 \left\| \frac{\mathbf{D}\boldsymbol{\delta}}{\mathbf{D}\mathbf{w}}(\tilde{\mathbf{w}}) \right\|_2 d\gamma, \tag{6}$$

*where* $\frac{\mathbf{D}\boldsymbol{\delta}}{\mathbf{D}\mathbf{w}}$ *denotes the Jacobian of optimal recourse with the corresponding operator matrix norm,* $\tilde{\mathbf{w}} := \gamma \mathbf{w} + (1 - \gamma)\mathbf{w}_{-i}$ *with* $\mathbf{w}_{-i}$ *being the optimal model parameters with the* $i$-th *training instance removed from the training set, and* $\mathbf{d}_i = \mathbf{w} - \mathbf{w}_{-i}$.

The norm of the Jacobian of optimal recourse indicates the local sensitivity of optimal recourse with respect to changes in model parameters $\mathbf{w}$. High magnitudes indicate that a small change in the parameters may require a fundamentally different recourse action. The total change can be bounded by the integral over these local sensitivities, which means that low local sensitivities along the path will result in a low overall change. Next, we specialize this result to the case of linear models.

**Corollary 1** (Upper bound on recourse action instability for linear models). *For the linear model* $f(\mathbf{x}) = \mathbf{w}_L^\top \mathbf{x}$ *with model parameters* $\mathbf{w}_L = (\mathbf{X}^\top \mathbf{X})^{-1}\mathbf{X}^\top \mathbf{Y}$, *an upper bound for the recourse action instability when* $s = 0, \lambda \to 0$ *by removing an instance from the training set is given by:*

$$\Phi_{\mathbf{x}}^{(2)} \leq \left( \max_{i \in [n]} \|\mathbf{d}_i^L\|_2 \right) \frac{4\sqrt{2}\|\mathbf{x}\|_2}{\min(\|\mathbf{w}_L\|_2, \min_{i \in [n]}\|\mathbf{w}_{L,-i}\|_2)}, \tag{7}$$

*under the condition that* $\mathbf{w}_L^\top \mathbf{w}_{L,-i} \geq 0$ *(no diametrical weight changes), where* $\mathbf{w}_{L,-i} = \mathbf{w}_L - \mathbf{d}_i^L$ *is the weight after removal of training instance* $i$ *and* $\mathbf{d}_i^L = (\mathbf{X}^T \mathbf{X})^{-1}\mathbf{x}_i \frac{(y_i - \mathbf{w}_L^\top \mathbf{x}_i)}{1 - h_{ii}}$.

For models trained on large data sets, the absolute value of the model parameters' norm $\|\mathbf{w}_L\|$ will not change much under deletion of a single instance. Therefore we argue that the denominator $\min(\|\mathbf{w_L}\|_2, \min_{i \in [n]}\|\mathbf{w}_{L,-i}\|_2) \approx \|\mathbf{w_L}\|$. Thus, recourse action instability is mainly determined by the sensitivity of model parameters to deletion, $\max_{i \in [n]} \|\mathbf{d}_i^L\|_2$, scaled by the ratio of $\frac{\|\mathbf{x}\|_2}{\|\mathbf{w_L}\|_2}$.

**Neural Tangent Kernels.** Studying the relation between deletion requests and the robustness of algorithmic recourse for models as complex as neural networks requires recent results from computational learning theory. In particular, we also rely on insights on the behaviour of over-parameterized neural networks from the theory of Neural Tangent Kernels (NTKs), which we will now briefly introduce. Thus we study our robustness notions for neural network models in the overparameterized regime with ReLU activation functions that take the following form:

$$f_{\text{ANN}}(\mathbf{x}) = \frac{1}{\sqrt{k}} \sum_{j=1}^{k} a_j \cdot \text{relu}(\mathbf{w}_j^\top \mathbf{x}), \tag{8}$$

where $\mathbf{W} = [\mathbf{w}_1, \ldots, \mathbf{w}_k] \in \mathbb{R}^{d \times k}$ and $\mathbf{a} = [a_1, \ldots, a_k] \in \mathbb{R}^k$. To concretely study the impact of data deletion on recourses in non-linear models such as neural networks, we leverage ideas from the neural tangent kernel (NTK) literature (Jacot et al., 2018; Lee et al., 2019; Arora et al., 2019; Du et al., 2019). The key insight from this literature for the purpose of our work is that infinitely wide neural networks can be expressed as a kernel ridge regression problem with the NTK under appropriate parameter initialization, and gradient descent training dynamics. In particular, in the limit as the number of hidden nodes $k \to \infty$, the neural tangent kernel associated with a two-layer ReLU network has a closed-form expression (Chen & Xu, 2021; Zhang & Zhang, 2021) (see Appendix A.5):

$$K^{\infty}(\mathbf{x}_0, \mathbf{x}) = \frac{\mathbf{x}_0^{\top} \mathbf{x} \left( \pi - \arccos \left( \frac{\mathbf{x}_0^{\top} \mathbf{x}}{\|\mathbf{x}_0\| \|\mathbf{x}\|} \right) \right)}{2\pi}. \tag{9}$$

Thus, the network's prediction at an input $\mathbf{x}$ can be described by:

$$f_{\text{NTK}}(\mathbf{x}) = \left( \mathbf{K}^{\infty}(\mathbf{x}, \mathbf{X}) \right)^{\top} \mathbf{w}_{\text{NTK}}, \tag{10}$$

where $\mathbf{X} \in \mathbb{R}^{n \times d}$ is the input data matrix, $\mathbf{K}^{\infty}(\mathbf{X}, \mathbf{X}) \in \mathbb{R}^{n \times n}$ is the NTK matrix evaluated on the training data points: $[\mathbf{K}^{\infty}(\mathbf{X}, \mathbf{X})]_{ij} = K^{\infty}(\mathbf{x}_i, \mathbf{x}_j)$ and $\mathbf{w}_{\text{NTK}} = \left( \mathbf{K}^{\infty}(\mathbf{X}, \mathbf{X}) + \beta \mathbf{I}_n \right)^{-1} \mathbf{Y}$ solves the $\ell_2$ regularized minimization problem with MSE loss where $\mathbf{Y} \in \mathbb{R}^n$ are the prediction targets. With this appropriate terminology in place we provide an upper bound on recourse outcome instability of wide neural network models.

**Proposition 3** (Upper bound on recourse outcome instability for wide neural networks)**.** *For the NTK model with $\mathbf{w}_{NTK} = \left( \mathbf{K}^{\infty}(\mathbf{X}, \mathbf{X}) + \beta \mathbf{I}_n \right)^{-1} \mathbf{Y}$, an upper bound for the recourse invalidation from Definition 1 by removing an instance $(\mathbf{x}, y)$ from the training set is given by:*

$$\Delta_{\mathbf{x}} \leq \|\mathbf{K}^{\infty}(\check{\mathbf{x}}_1, \mathbf{X})\|_2 \cdot \max_{i \in [n]} \|\mathbf{d}_i^{NTK}\|_2, \tag{11}$$

*where $\mathbf{d}_i^{NTK} = \frac{1}{k_{ii}} \mathbf{k}_i \mathbf{k}_i^{\top} \mathbf{Y}$, where $\mathbf{k}_i$ is the $i$-th column of the matrix $\left( \mathbf{K}^{\infty}(\mathbf{X}, \mathbf{X}) + \beta \mathbf{I}_n \right)^{-1}$, and $k_{ii}$ is its $i$-th diagonal element.*

Intuitively, $\mathbf{d}_i^{\text{NTK}}$ is the linear model analog to $\mathbf{d}_i^{\text{L}}$ and $\mathbf{d}_i^{\text{NTK}}$ represents the importance that the point $\mathbf{z}_i = (\mathbf{x}_i, y_i)$ has on the model parameters $\mathbf{w}_{\text{NTK}}$.

In practical use-cases, when trying to comply with both the right to data deletion and the right to actionable explanations, our results have practical implications. For example, instances with high influence captured by $\mathbf{d}_i$ should be encountered with caution during model training in order to provide reliable recourses to the individuals who seek recourse. In summary, our results suggest that the right to data deletion may be fundamentally at odds with reliable state-of-the-art actionable explanations as the removal of an influential data instance can induce a large change in the recourse robustness, the extent to which is primarily measured by the empirical influence function $\mathbf{d}_i$.

## 5    Finding the Set of Most Critical Data Points

**The Objective Function.** In this section, we present optimization procedures that can be readily used to assess recourses' vulnerability to deletion requests. On this way, we start by formulating our optimization objective. We denote by $m \in \{\Delta, \Phi^{(2)}\}$ the measure we want to optimize for. We consider the summed instability of over the data set by omitting the subscript $\mathbf{x}$, e.g., $\Delta = \sum_{\mathbf{x} \in \mathcal{D}_{test}} \Delta_{\mathbf{x}}$. Our goal is to find the smallest number of deletion requests that leads to a maximum impact on the instability measure $m$. To formalize this, define the set of data weight configurations:

$$\Gamma_{\alpha} := \{\boldsymbol{\omega} : \text{Maximally } \lfloor \alpha \cdot n \rfloor \text{ entries of } \boldsymbol{\omega} \text{ are } 0 \text{ and the remainder is } 1.\}. \tag{12}$$

In equation 12, the parameter $\alpha$ controls the fraction of instances that are being removed from the training set. For a fixed fraction $\alpha$, our problem of interest becomes:

$$\boldsymbol{\omega}^* = \arg\max_{\boldsymbol{\omega} \in \Gamma_{\alpha}} m(\boldsymbol{\omega}). \tag{13}$$

**Fundamental Problems.** When optimizing the above objective we face two fundamental problems: (i) *evaluating $m(\boldsymbol{\omega})$ for many weight configurations $\boldsymbol{\omega}$ can be prohibitively expensive as the objective*

is defined implicitly through solutions of several non-linear optimization problems (i.e., model fitting and finding recourses). Further, (ii) even for an objective $m(\boldsymbol{\omega})$ which can be computed in constant or polynomial time *optimizing* this objective can still be NP-hard (a proof is given in Appendix A.3).

**Practical Algorithms.** We devise two practical algorithms which approach the problem in equation 13 in different ways. As for the problem of computing $m(\boldsymbol{\omega})$ in (i), we can either solve this by (a) using a closed-form expression indicating the dependency of $m$ on $\boldsymbol{\omega}$ or (b) by using an approximation of $m$ that is differentiable with respect to $\boldsymbol{\omega}$. As for the optimization in (ii), once we have established the dependency of $m$ on $\boldsymbol{\omega}$ we can either (a) use a gradient descent approach or (b) we use a greedy method. Below we explain the individual steps required for our algorithms.

## 5.1 COMPUTING THE OBJECTIVE

In the objective $m(\boldsymbol{\omega})$, notice the dependencies $\Delta_{\mathbf{x}}(\boldsymbol{\omega}) = \Delta_{\mathbf{x}}(f(\mathbf{w}(\boldsymbol{\omega}), \check{\mathbf{x}}))$ for the recourse outcome instability, and $\Phi_{\mathbf{x}}^{(2)}(\boldsymbol{\omega}) = \Phi_{\mathbf{x}}^{(2)}(\boldsymbol{\delta}(\mathbf{w}(\boldsymbol{\omega}), \mathbf{x})))$ for the recourse action instability. In the following, we briefly discuss how we efficiently compute each of these functions without numerical optimization.

**Model parameters from data weights $\mathbf{w}(\boldsymbol{\omega})$.** For the linear model, an analytical solution can be obtained, $\mathbf{w}_{\mathrm{L}}(\boldsymbol{\omega}) = (\mathbf{X}^{\top}\boldsymbol{\Omega}\mathbf{X})^{-1}\mathbf{X}^{\top}\boldsymbol{\Omega}\mathbf{Y}$, where $\boldsymbol{\Omega} = \mathrm{diag}(\boldsymbol{\omega})$. The same goes for the NTK model where $\mathbf{w}_{\mathrm{NTK}}(\boldsymbol{\omega}) = \boldsymbol{\Omega}^{\frac{1}{2}}(\boldsymbol{\Omega}^{\frac{1}{2}}\mathbf{K}^{\infty}(\mathbf{X}, \mathbf{X})\boldsymbol{\Omega}^{\frac{1}{2}} + \beta\mathbf{I})^{-1}\boldsymbol{\Omega}^{\frac{1}{2}}\mathbf{Y}$ (Busuttil & Kalnishkan, 2007, Eqn. 3). When no closed-form expressions for the model parameters exist, we can resort to the infinitesimal jackknife (IJ) (Jaeckel, 1972; Efron, 1982; Giordano et al., 2019b;a), that can be seen as a linear approximation to this implicit function. We refer to Appendix C for additional details on this matter.

**Model prediction from model parameters $f(\mathbf{w}, \check{\mathbf{x}})$.** Having established the model parameters, evaluating the prediction at a given point can be quickly done even in a differentiable manner with respect to $\mathbf{w}$ for the models we consider in this work.

**Recourse action from model parameters $\boldsymbol{\delta}(\mathbf{w}, \check{\mathbf{x}})$.** Estimating the recourse action is more challenging as it requires solving equation 2. However, a differentiable solution exists for linear models, where the optimal recourse action is given by $\boldsymbol{\delta}_{\mathrm{L}} = \frac{s - \mathbf{w}_{\mathrm{L}}(\boldsymbol{\omega})^{\top}\mathbf{x}}{\lambda + \|\mathbf{w}_{\mathrm{L}}(\boldsymbol{\omega})\|_2^2}\mathbf{w}_{\mathrm{L}}(\boldsymbol{\omega})$. When the underlying predictor is a wide neural network we can approximate the recourse expression of the corresponding NTK, $\boldsymbol{\delta}_{\mathrm{NTK}} \approx \frac{s - f_{\boldsymbol{\omega},\mathrm{NTK}}(\mathbf{x})}{\lambda + \|\bar{\mathbf{w}}_{\mathrm{NTK}}(\boldsymbol{\omega})\|_2^2}\bar{\mathbf{w}}_{\mathrm{NTK}}(\boldsymbol{\omega})$, which stems from the first-order taylor expansion $f_{\boldsymbol{\omega},\mathrm{NTK}}(\mathbf{x} + \boldsymbol{\delta}) \approx f_{\boldsymbol{\omega},\mathrm{NTK}}(\mathbf{x}) + \boldsymbol{\delta}^{\top}\bar{\mathbf{w}}_{\mathrm{NTK}}(\boldsymbol{\omega})$ with $\bar{\mathbf{w}}_{\mathrm{NTK}}(\boldsymbol{\omega}) = \nabla_{\mathbf{x}}K(\mathbf{x}, \mathbf{X})\mathbf{w}_{\mathrm{NTK}}(\boldsymbol{\omega})$.

## 5.2 OPTIMIZING THE OBJECTIVE FUNCTION

**The Greedy Algorithm.** We consider the model on the full data set and compute the objective function $m(\boldsymbol{\omega})$ under deletion of every instance (alone). We then select the instance that leads to the highest increase in the objective. We add this instance to the set of deleted points. Subsequently, we refit the model and compute the impact of deletion for every second instance, when deleted in combination with the first one. Again, we add the instance that results in the largest increase to the set. Iteratively repeating these steps, we identify more instances to be deleted. Computational complexity depends on the implementation of the model weight recomputation, which is required $\mathcal{O}(\alpha n^2)$ times.

**The Gradient Descent Algorithm.** Because our developed computation of $m(\boldsymbol{\omega})$ can be made differentiable, we also propose a *gradient-based optimization* framework. We consider the relaxation of the problem in equation 13,

$$\boldsymbol{\omega}^* = \arg\max_{\boldsymbol{\omega} \in \{0,1\}^n} m(\boldsymbol{\omega}) - \|\mathbf{1} - \boldsymbol{\omega}\|_0, \tag{14}$$

where the $\ell_0$ norm encourages to change as few data weights from $1$ to $0$ as possible while few removals of training instances should have maximum impact on the robustness measure. The problem in equation 14 can be further relaxed to a continuous and unconstrained optimization problem. To do so we use a recently suggested stochastic surrogate loss for the $\ell_0$ term (Yamada et al., 2020). Using this technique, a surrogate loss for equation 14 can be optimized using stochastic gradient descent (SGD). We refer to Appendix C for more details and pseudo-code of the two algorithms.

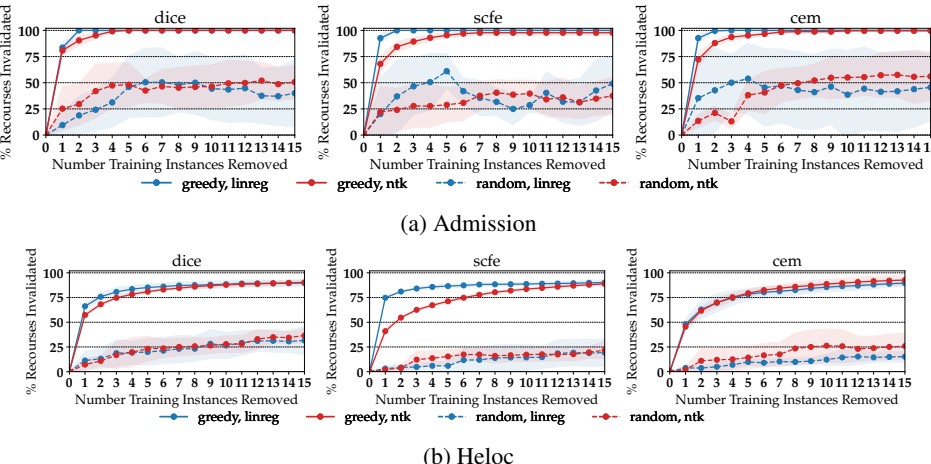

(a) Admission

(b) Heloc

Figure 2: Measuring the tradeoff between recourse outcome instability and the number of deletion requests for both the Admission and the Heloc data sets for regression and NTK models and various recourse methods. Results were obtained by greedy optimization; see Appendix B for SGD results.

## 6 EXPERIMENTAL EVALUATION

We experimentally evaluate our framework in terms of its ability to find significant recourse invalidations using the instability measures presented in Section 3.

**Data Sets.** For our experiments on regression tasks we use two real-world data sets. In addition, we provide results for two classification datasets in the Appendix B. First, we use law school data from the Law School Admission Council (*Admission*). The council carried out surveys across 163 law schools in the US, in which they collected information from 21,790 law students across the US (Wightman, 1998). The data contains information on the students' prior performances. The task is to predict the students' first-year law-school average grades. Second, we use the *Home Equity Line of Credit (Heloc)* data set. Here, the target variable is a score indicating whether individuals will repay the Heloc account within a fixed time window. Across both tasks we consider individuals in need of recourse if their scores lie below the median score across the data set.

**Recourse Methods.** We apply our techniques to four different methods which aim to generate low-cost recourses using different principles: SCFE was suggested by Wachter et al. (Wachter et al., 2018) and uses a gradient-based objective to find recourses, DICE (Mothilal et al., 2020) uses a gradient-based objective to find recourses subject to a diversity constraint, and CEM (Dhurandhar et al., 2018) uses a generative model to encourage recourses to lie on the data manifold. For all methods, we used the recourse method implementations from the CARLA library (Pawelczyk et al., 2021) and specify the $\ell_1$ cost constraint. Further details on these algorithms are provided in App. C.

**Evaluation Measures.** For the purpose of our evaluation, we use both the recourse outcome instability measure and the recourse action instability measure presented in Definitions 1 and 2. We evaluate the efficacy of our framework to destabilize a large fraction of recourses using a small number of deletion requests (up to 14). To find critical instances, we use the greedy and the gradient-based algorithms described in Sec. 5. After having established a set of critical points, we recompute the metrics with the refitted models and recourses to obtain a ground truth result.

For the *recourse outcome instability*, our metric $\Delta$ counts the number of invalidated recourses. We use the median as the target score $s$, i.e., if the recourse outcome flips back from a positive leaning prediction (above median) to a negative one (below median) it is considered invalidated. When evaluating *recourse action instability*, we identify a set of critical points, delete these points from the train set and refit the predictive model. In this case, we also have to recompute the recourses to evaluate $\Phi_p$. We then measure the recourse instability using Definition 2 with $p = 2$. Additionally, we compare with a random baseline, which deletes points uniformly at random from the train set. We compute these measures for all individuals from the test set who require algorithmic recourse. To obtain standard errors, we split the test set into 5 folds and report averaged results over these 5 folds.

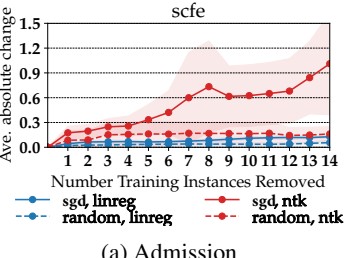

(a) Admission

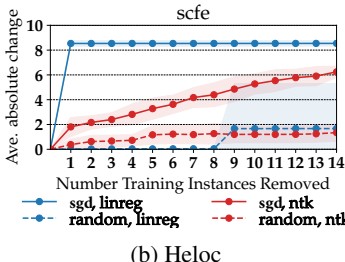

(b) Heloc

Figure 3: Quantifying the tradeoff between recourse action instability as measured in Definition 2 and the number of deletion requests for both the Admission and the Heloc data sets for the SCFE method when the underlying model is linear or an NTK (results by SGD optimization).

**Results.** In Figure 2, we measure the tradeoff between *recourse outcome instability* and the number of deletion requests. We plot the number of deletion requests against the fraction of all recourses that become invalidated when up to $k \in \{1, \ldots, 14\}$ training points are removed from the training set of the predictive model. When the underlying model is linear, we observe that the removal of as few as 5 training points induces invalidation rates of all recourses that are as high as 95 % percent – we observe a similar trend across all recourse methods. Note that a similar trend is present for the NTK model; however, a larger number of deletion requests (roughly 9) is required to achieve similar invalidation rates. Finally, also note that our approach is always much more effective at deleting instances than the random baseline. In Figure 3, we measure the tradeoff between *recourse action instability* and the number of deletion requests with respect to the SCFE recourse method when the underlying predictive model is linear or an NTK model. For this complex objective, we use the more efficient SGD optimization. Again, we observe that our optimization method significantly outperforms the random baselines at finding the most influential points to be removed.

**Additional Models and Tasks.** In addition to the here presented results, we provide results for classification tasks with (a) Logistic Regression, (b) Kernel-SVM and (c) ANN models on two additional data sets (Diabetes and COMPAS) in Appendix B. Across all these models, we observe that our removal algorithms outperform random guessing; often by up to 75 percentage points.

**Factors of Recourse Robustness.** Our empirical results shed light on which factors are influential in determining robustness of trained ML models with respect to deletion requests. In combination with results from Fig. 4 (see Appendix B), our results suggest that linear models are more susceptible to invalidation in the worst-case but are slightly more robust when it comes to random removals. Furthermore, the characteristics of the data set play a key role; in particular those of the critical points. We perform an additional experiment where we consider modified data sets without the most influential points identified by our optimization approaches. In Appendix B, initial results show that this simple technique decreases the invalidation probabilities by up to 6 percentage points.

## 7 DISCUSSION AND CONCLUSION

In this work, we made the first step towards understanding the tradeoffs between actionable model explanations and the right to be forgotten. We theoretically analyzed the robustness of state-of-the-art recourse methods under data deletion requests and suggested (i) a greedy and (ii) a gradient-based algorithm to efficiently identify a small subset of individuals, whose data, when removed, would lead to invalidation of a large number of recourses for unsuspecting other users. Our experimental evaluation with multiple real-world data sets on both regression and classification tasks demonstrates that the right to be forgotten presents a significant challenge to the reliability of actionable explanations.

Finally, our theoretical results suggest that the robustness to deletion increases when the model parameter changes under data deletion remain small. This formulation closely resembles the definition of *Differential Privacy* (DP) (Dwork et al., 2014). We therefore conjecture that the reliability of actionable recourse could benefit from models that have been trained under DP constraints. As the field of AI rapidly evolves, data protection authorities will further refine the precise interpretations of general principles in regulations such as GDPR. The present paper contributes towards this goal theoretically, algorithmically, and empirically by providing evidence of tensions between different data protection principles.

**Ethics statement.** Our findings raise compelling questions on the deployment of counterfactual explanations in practice. First of all, *Are the two requirements of actionable explanations and the right to be forgotten fundamentally at odds with one another?* The theoretical and empirical results in this work indicate that for many model and recourse method pairs, this might indeed be the case. This finding leads to the pressing follow-up question: *How can practitioners make sure that their recourses stay valid under deletion requests?* A first take might be to implement the principle of data minimization (Biega et al., 2020; Biega & Finck, 2021; Shanmugam et al., 2022) in the first place, i.e., exclude the $k$ most critical data points from model training.

**Acknowledgments.** We would like to thank the anonymous reviewers for their insightful feedback. MP would like to thank Jason Long, Emanuele Albini, Jiāháo Chén, Daniel Dervovic and Daniele Magazzeni for insightful discussions at early stages of this work.

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

APPENDIX

# A  THEORETICAL RESULTS

## A.1  UPPER BOUNDS ON RECOURSE OUTCOME INSTABILITY

**Proposition 1** (Upper Bound on Output Robustness for Linear Models). *For the linear regression model $f(\mathbf{x}) = \mathbf{w}^\top \mathbf{x}$ with weights given by $\mathbf{w} = (\mathbf{X}^\top \mathbf{X})^{-1} \mathbf{X}^\top \mathbf{Y}$, an upper bound for the output robustness by removing an instance $(\mathbf{x}, y)$ from the training set is given by:*

$$\Delta_{\mathbf{x}} \leq \max_{i \in [n]} \|\mathbf{d}_i\|_2 \cdot \|\check{\mathbf{x}}_1\|_2, \tag{15}$$

*where $\mathbf{d}_i = (\mathbf{X}^\top \mathbf{X})^{-1} \mathbf{x}_i \cdot \frac{r_i}{1-h_{ii}}$, $r_i = y_i - \mathbf{w}^\top \mathbf{x}_i$ and $h_{ii} = \mathbf{x}_i^\top (\mathbf{X}^\top \mathbf{X})^{-1} \mathbf{x}_i$.*

*Proof.* By Definition 1, we have:

$$\Delta_{\mathbf{x}} = \left| \mathbf{w}_1^\top \check{\mathbf{x}}_1 - \mathbf{w}_{-i}^\top \check{\mathbf{x}}_1 \right| \tag{16}$$

$$= \left| \left( \mathbf{w}_1 - \mathbf{w}_{-i} \right)^\top \check{\mathbf{x}}_1 \right|$$

$$= \left| \left( (\mathbf{X}^T \mathbf{X})^{-1} \mathbf{x}_i \frac{(y_i - \mathbf{w}^T \mathbf{x}_i)}{1 - h_{ii}} \right)^\top \check{\mathbf{x}}_1 \right| \qquad \text{(by Theorem 1)} \tag{17}$$

$$\leq \|\mathbf{d}_i\|_2 \cdot \|\check{\mathbf{x}}_1\|_2 \qquad \text{(by Cauchy-Schwartz)} \tag{18}$$

$$\leq \|\check{\mathbf{x}}_1\|_2 \cdot \max_{i \in [n]} \|\mathbf{d}_i\|_2 \cdot, \tag{19}$$

where $\mathbf{d}_i = (\mathbf{X}^T \mathbf{X})^{-1} \mathbf{x}_i \frac{(y_i - \mathbf{w}^T \mathbf{x}_i)}{1 - h_{ii}}$. This completes our proof. $\qquad\square$

**Proposition 2** (Upper Bound on Output Robustness for NTK). *For the NTK model with $\mathbf{w}_{NTK} = \left( \mathbf{K}^\infty(\mathbf{X}, \mathbf{X}) + \lambda \mathbf{I}_n \right)^{-1} \mathbf{Y}$, an upper bound for the output robustness by removing an instance $(\mathbf{x}, y)$ from the training set is given by:*

$$\Delta_{\mathbf{x}} \leq \|\mathbf{K}^\infty(\check{\mathbf{x}}_1, \mathbf{X})\|_2 \cdot \max_{i \in [n]} \|\mathbf{d}_i\|_2, \tag{20}$$

*where $\mathbf{d}_i = \frac{1}{k_{ii}} \mathbf{k}_i \mathbf{k}_i^\top \mathbf{Y}$, where $\mathbf{k}_i$ is the $i$-th column of the matrix $\left( \mathbf{K}^\infty(\mathbf{X}, \mathbf{X}) + \beta \mathbf{I}_n \right)^{-1}$, and $k_{ii}$ is its $i$-th diagonal element.*

*Proof.* By Definition 1 and the weight-update theorem by Zhang & Zhang (2021) (see Appendix A.4) and the assumption of the over-parameterized regime, we have:

$$\Delta_{\mathbf{x}} = \left| f_{\text{NTK}}(\check{\mathbf{x}}_1) - f_{\text{NTK}}^{-i}(\check{\mathbf{x}}_1) \right|$$

$$= \left| \left( \mathbf{K}^\infty(\check{\mathbf{x}}_1, \mathbf{X}) \right)^\top \mathbf{w}_{\text{NTK}} - \left( \mathbf{K}^\infty(\check{\mathbf{x}}_1, \mathbf{X}) \right)^\top \left( \left( \mathbf{K}^\infty(\mathbf{X}, \mathbf{X}) + \beta \mathbf{I}_n \right)^{-1} - \frac{1}{k_{ii}} \mathbf{k}_i \mathbf{k}_i^\top \right) \mathbf{Y} \right| \tag{21}$$

$$= \left| \left( \mathbf{K}^\infty(\check{\mathbf{x}}_1, \mathbf{X}) \right)^\top \frac{1}{k_{ii}} \mathbf{k}_i \mathbf{k}_i^\top \mathbf{Y} \right| \tag{22}$$

$$\leq \|\mathbf{d}_i\|_2 \cdot \|\mathbf{K}^\infty(\check{\mathbf{x}}_1, \mathbf{X})\|_2 \quad \text{(by Cauchy-Schwartz)}$$

$$\leq \|\check{\mathbf{x}}_1\|_2 \cdot \max_{i \in [n]} \|\mathbf{d}_i\|_2, \tag{23}$$

where $\mathbf{d}_i = \frac{1}{k_{ii}} \mathbf{k}_i \mathbf{k}_i^\top \mathbf{Y}$ which completes our proof. $\qquad\square$

## A.2  UPPER BOUNDS ON RECOURSE ACTION INSTABILITY

**Proposition 3** (Upper Bound on Input Robustness). *For the linear regression model $f(\mathbf{x}) = \mathbf{w}^\top \mathbf{x}$ with weights given by $\mathbf{w} = (\mathbf{X}^\top \mathbf{X})^{-1} \mathbf{X}^\top \mathbf{Y}$, an upper bound for the input robustness in the setting $s = 0, \lambda = 0$ by removing the $i$-th instance $(\mathbf{x}_i, y_i)$ from the training set is given by:*

$$\Phi_{\mathbf{x}}^{(2)} \leq \|\mathbf{d}_i\|_2 \frac{4\sqrt{2}\|\mathbf{x}\|_2}{\min(\|\mathbf{w}\|_2, \|\mathbf{w}_{-i}\|_2)}, \tag{24}$$

*under the condition that $\mathbf{w}^\top \mathbf{w}_{-i} \leq 0$ (no diametrical weight changes), where $\mathbf{w}_{-i} = \mathbf{w} - \mathbf{d}_i$ is the weight after removal of training instance $i$ and $\mathbf{d}_i = (\mathbf{X}^T\mathbf{X})^{-1}\mathbf{x}_i \frac{(y_i - \mathbf{w}^\top \mathbf{x}_i)}{1 - h_{ii}}$.*

*Proof.* For a linear scoring function $f(\mathbf{x}) = \mathbf{w}'^\top \mathbf{x}$ with given parameters $\mathbf{w}'$, under the squared $\ell_2$ norm constraint with balance parameter $\lambda$, the optimal recourse action is given by (Pawelczyk et al., 2022):

$$\boldsymbol{\delta}(\mathbf{w}') = \frac{s - \mathbf{w}'^\top \mathbf{x}}{\|\mathbf{w}'\|_2^2 + \lambda} \cdot \mathbf{w}'. \tag{25}$$

Using Definition 2, we can express the total change in $\boldsymbol{\delta}$ as a path integral over changes in $\mathbf{w}$, times the change $\frac{\mathbf{D}\boldsymbol{\delta}}{\mathbf{D}\mathbf{w}}$ they entail:

$$\Phi_{\mathbf{x}}^{(2)} = \left\|\boldsymbol{\delta_1} - \boldsymbol{\delta_\omega}\right\|_2 = \left\|\boldsymbol{\delta}(\mathbf{w}) - \boldsymbol{\delta}(\mathbf{w}_{-i})\right\|_2 \tag{26}$$

$$\leq \int_0^1 \left\|\frac{\mathbf{D}\boldsymbol{\delta}}{\mathbf{D}\mathbf{w}}(\gamma\mathbf{w} + (1-\gamma)\mathbf{w}_{-i})\right\| \|\mathbf{w} - \mathbf{w}_{-i}\|_2 d\gamma, \tag{27}$$

where $\frac{\mathbf{D}\boldsymbol{\delta}}{\mathbf{D}\mathbf{w}}$ denotes the Jacobian, with the corresponding operator matrix norm. Defining $\tilde{\mathbf{w}} := \gamma\mathbf{w} + (1-\gamma)\mathbf{w}_{-i}$ and using $\|\mathbf{w} - \mathbf{w}_{-i}\|_2 = \|\mathbf{d}_i\|_2$, we obtain

$$\Phi_{\mathbf{x}}^{(2)} \leq \|\mathbf{d}_i\|_2 \int_0^1 \left\|\frac{\mathbf{D}\boldsymbol{\delta}}{\mathbf{D}\mathbf{w}}(\tilde{\mathbf{w}})\right\|_2 d\gamma. \tag{28}$$

Because of the form $\boldsymbol{\delta}(\mathbf{w}') = f(\mathbf{w}')\mathbf{w}'$, where $f(\mathbf{w}') := \frac{s - \mathbf{w}'^\top \mathbf{x}}{\|\mathbf{w}'\|_2^2 + \lambda}$ is a scalar function, its Jacobian has the form $\frac{\mathbf{D}\boldsymbol{\delta}}{\mathbf{D}\mathbf{w}'} = \mathbf{w}'(\nabla f(\mathbf{w}'))^\top + f(\mathbf{w}')\mathbf{I}$. We will now derive a bound on the Jacobian's operator norm:

$$\left\|\frac{\mathbf{D}\boldsymbol{\delta}}{\mathbf{D}\mathbf{w}'}(\tilde{\mathbf{w}})\right\|_2 = \max_{\|\mathbf{a}\|=1} \left\|\frac{\mathbf{D}\boldsymbol{\delta}}{\mathbf{D}\mathbf{w}'}\mathbf{a}\right\|_2 = \max_{\|\mathbf{a}\|=1} \left\|\mathbf{w}'(\nabla f(\tilde{\mathbf{w}}))^\top \mathbf{a} + f(\tilde{\mathbf{w}})\mathbf{a}\right\|_2 \tag{29}$$

$$\leq \|\nabla f(\tilde{\mathbf{w}})\|_2 \|\tilde{\mathbf{w}}\|_2 + |f(\tilde{\mathbf{w}})|. \tag{30}$$

Additionally, we know that for $s = 0$, $|f(\mathbf{w}')| \leq \frac{\|\mathbf{x}\|_2 \|\tilde{\mathbf{w}}\|_2}{\|\tilde{\mathbf{w}}\|_2^2} = \frac{\|\mathbf{x}\|_2}{\|\tilde{\mathbf{w}}\|_2}$. The gradient is given by

$$\|\nabla f(\tilde{\mathbf{w}})\|_2 = \left\|\frac{-(\|\tilde{\mathbf{w}}\|_2^2 + \lambda)\mathbf{x} - 2(s - \tilde{\mathbf{w}}^\top \mathbf{x})\tilde{\mathbf{w}}}{(\|\tilde{\mathbf{w}}\|_2^2 + \lambda)^2}\right\|_2 \tag{31}$$

$$\leq \frac{(\|\tilde{\mathbf{w}}\|_2^2 + \lambda)\|\mathbf{x}\|_2 + 2(s + \|\tilde{\mathbf{w}}\|_2\|\mathbf{x}\|_2)\|\tilde{\mathbf{w}}\|_2}{\|\tilde{\mathbf{w}}\|_2^4} \tag{32}$$

$$= \frac{3\|\mathbf{x}\|_2}{\|\tilde{\mathbf{w}}\|_2^2} \qquad \text{(Using } \lambda \to 0, s = 0\text{)}. \tag{33}$$

In summary,

$$\left\|\frac{\mathbf{D}\boldsymbol{\delta}}{\mathbf{D}\mathbf{w}'}(\tilde{\mathbf{w}})\right\|_2 \leq \frac{3\|\mathbf{x}\|_2}{\|\tilde{\mathbf{w}}\|_2^2}\|\tilde{\mathbf{w}}\|_2 + \frac{\|\mathbf{x}\|_2}{\|\tilde{\mathbf{w}}\|_2} = \frac{4\|\mathbf{x}\|_2}{\|\tilde{\mathbf{w}}\|_2}. \tag{34}$$

Because $\tilde{\mathbf{w}}$ is a line between $\mathbf{w}$ and $\mathbf{w}_{-i}$, its norm is bounded from below by $\|\tilde{\mathbf{w}}\|_2 \geq \frac{1}{\sqrt{2}}\min(\|\mathbf{w}\|_2, \|\mathbf{w}_{-i}\|_2) \geq \frac{1}{\sqrt{2}}(\|\mathbf{w}\|_2 - \|\mathbf{w} - \mathbf{w}_{-i}\|_2) = \frac{1}{\sqrt{2}}(\|\mathbf{w}\|_2 - \|\mathbf{d}_i\|_2)$. We can thus uniformly bound the integral and plug in the bound because of its positivity,

$$\Phi_{\mathbf{x}}^{(2)} \leq \|\mathbf{d}_i\|_2 \int_0^1 \left\|\frac{\mathbf{D}\boldsymbol{\delta}}{\mathbf{D}\mathbf{w}}(\tilde{\mathbf{w}})\right\|_2 d\gamma \tag{35}$$

$$\leq \|\mathbf{d}_i\|_2 \int_0^1 \frac{4\sqrt{2}\|\mathbf{x}\|_2}{\min(\|\mathbf{w}\|_2, \|\mathbf{w}_{-i}\|_2)} d\gamma \tag{36}$$

$$= \|\mathbf{d}_i\|_2 \frac{4\sqrt{2}\|\mathbf{x}\|_2}{\min(\|\mathbf{w}\|_2, \|\mathbf{w}_{-i}\|_2)}, \tag{37}$$

which completes the proof. $\qquad\square$

A.3 CALCULATING RECOURSE OUTCOME INSTABILITY FOR k DELETIONS IS NP-HARD

We can show that, for a general scoring function $f$, the problem defined in equation 13 is NP-hard. We make this proof by providing a function $f$ for which solving the recourse outcome invalidity problem is as hard as solving the well-known Knapsack problem, that has been shown to be NP-hard (Karp, 1972). The knapsack problem is defined as follows:

$$\max_{q_i \in \{0,1\}} \sum_{i=1}^{n} v_i q_i \text{ s.t. } \sum_{i=1}^{n} y_i q_i \leq W, \tag{38}$$

where the problem considers $n$ fixed items $(v_i, y_i)_{i=1...n}$ with a value $v_i$ and knapsack weight $y_i > 0$, and $W$ is a fixed weight budget. The optimization problem consists of choosing the items that maximize the summed values but have a weight lower than $W$. To solve this problem through the recourse outcome invalidation problem, we suppose there is a data point for each item. We can choose any $k > \frac{W}{\min y_i}$ of points to be deleted, where this condition ensures that we can remove the number of samples maximally required to solve the corresponding knapsack problem. Note that we can always add a number of dummy points that have no effect such that the total number of data points is at least $k$. Suppose there is a classifier function:

$$f_{\boldsymbol{\omega}}(\mathbf{x}) := \begin{cases} \sum_{i=1}^{n} v_i(1 - \omega_i), & \sum_{i=1}^{n} y_i(1 - \omega_i) \leq W \\ 0, & \text{else} \end{cases} . \tag{39}$$

In this case, solving Eqn. 13 comes down to finding the set of items (i.e., removing the data points) that have maximum value, but stay under the threshold $W$. Thus, if we can solve Eqn. 13, the solution to the equivalent knapsack problem is given by $\mathbf{q} = (\mathbf{1} - \boldsymbol{\omega})$.

A.4 AUXILIARY THEORETICAL RESULTS

We state the following classic result by Miller Jr (1974) without proof.

**Theorem 1.** *(Leave-One-Out Estimator, Miller Jr (1974)) Define $(\mathbf{x}_i, y_i)$ as the point to be removed from the training set. Given the optimal weight vector $\mathbf{w} = (\mathbf{X}^\top \mathbf{X})^{-1} \mathbf{X}^\top \mathbf{Y}$ which solves for a linear model under mean-squared-error loss, the leave-one-out estimator is given by:*

$$\mathbf{w} - \mathbf{w}_{-i} = (\mathbf{X}^T \mathbf{X})^{-1} \mathbf{x}_i \frac{(y_i - \mathbf{w}^T \mathbf{x}_i)}{1 - \mathbf{x}_i^T (\mathbf{X}^T \mathbf{X})^{-1} \mathbf{x}_i} = (\mathbf{X}^T \mathbf{X})^{-1} \mathbf{x}_i \frac{(y_i - \mathbf{w}^T \mathbf{x}_i)}{1 - h_{ii}} =: \mathbf{d}_i.$$

We restate the analtical solution for the NTK weights in case of a single deletion from Zhang & Zhang (2021).

**Theorem 2.** *(Leave-One-Out weights for NTK models, Zhang & Zhang (2021)) Let $\mathbf{w}_{NTK} = \left(\mathbf{K}^\infty(\mathbf{X}, \mathbf{X}) + \lambda \mathbf{I}_n\right)^{-1} \mathbf{Y}$ be the weight for the NTK model on the full data Kernel model, where $\mathbf{K}^\infty(\mathbf{X}, \mathbf{X})$ is the NTK matrix evaluated on the training data points: $[\mathbf{K}^\infty(\mathbf{X}, \mathbf{X})]_{ij} = K^\infty(\mathbf{x}_i, \mathbf{x}_j)$. Then, the NTK model that would be obtained when removing instance $i$, could be equivalently described by*

$$f_{NTK}^{-i}(\mathbf{x}) = \mathbf{K}^\infty(\mathbf{x}, \mathbf{X})^\top \mathbf{w}_{NTK,-i} = \mathbf{K}^\infty(\mathbf{x}, \mathbf{X})^\top \left( \left(\mathbf{K}^\infty(\mathbf{X}, \mathbf{X}) + \lambda \mathbf{I}_n\right)^{-1} - \frac{1}{q_{-ii}} \boldsymbol{q}_{-i} \boldsymbol{q}_{-i}^\top \right) \mathbf{Y}$$

*where $\boldsymbol{q}_{-i}$ is the $i$-th column of the matrix $\boldsymbol{Q}^{-1} = \left(\mathbf{K}^\infty(\mathbf{X}, \mathbf{X}) + \beta \mathbf{I}_n\right)^{-1}$, and $q_{-ii}$ is its $i$-th diagonal element of this inverse.*

*Proof.* We begin by introducing some notation. Define:

$$\boldsymbol{Q} = \mathbf{K}^\infty(\mathbf{X}, \mathbf{X}) + \beta \mathbf{I}_n \tag{40}$$
$$\boldsymbol{R} = \mathbf{K}^\infty(\mathbf{X}_{-i}, \mathbf{X}_{-i}) + \beta \mathbf{I}_{n-1}, \tag{41}$$

then the analytical NTK model when considering the dataset with one instance $\mathbf{x}_i$ removed is given by

$$f_{\text{NTK}}^{-i}(\mathbf{x}) = \mathbf{K}^\infty(\mathbf{x}, \mathbf{X}_{-i})^\top \boldsymbol{R}^{-1} \mathbf{Y}_{-i}, \tag{42}$$

where $\mathbf{X}_{-i}$ denotes the data matrix with row $i$ missing and $\mathbf{Y}_{-i}$ denotes the label vector with the $i$-th label missing. We have to show that this expression is equivalent to that stated in the theorem.

Without loss of generality, we can assume the $i$ is the last point in the dataset (otherwise, we just permute the data set accordingly). Therefore, we can write the matrix $Q$ in block form:

$$Q = \begin{bmatrix} R & \mathbf{K}^\infty(\mathbf{x}_i, X) \\ \mathbf{K}^{\infty\top}(\mathbf{x}_i, X) & \mathbf{K}^\infty(\mathbf{x}_i, \mathbf{x}_i) \end{bmatrix} := \begin{bmatrix} R & q_i \\ q_i^\top & q_{ii} \end{bmatrix} \tag{43}$$

Through the block matrix inversion formula (see for example Csató & Opper (2002) (eqn. 52)) we can write $Q$'s inverse as

$$Q^{-1} = \begin{bmatrix} R^{-1} + \gamma^{-1} R^{-1} q_i q_i^\top R^{-1} & \gamma^{-1} R^{-1} q_i \\ \gamma^{-1} q_i^\top R^{-1} & \gamma^{-1} \end{bmatrix} \tag{44}$$

with $\gamma = q_{ii} - q_i^\top R^{-1} q_i$.

We denote the $i$-th (and last) column of $Q^{-1}$ as $q_{-i} = \begin{bmatrix} \gamma^{-1} R^{-1} q_i \\ \gamma^{-1} \end{bmatrix}$ and the $i$-th and last diagonal element of the inverse as $q_{-ii} = \gamma^{-1}$. We will now show, that the form of the weights given in the theorem (i.e., the weights for the points not removed) are equivalent to the weights that would have been computed by plugging in the smaller kernel matrix $\mathbf{K}^\infty(\mathbf{X}_{-i}, \mathbf{X}_{-i})$ in the analytical solution and the weight for the point deleted will have a value of zero, i.e.,

$$\left( (\mathbf{K}^\infty(\mathbf{X}, \mathbf{X}) + \beta \mathbf{I}_n)^{-1} - \frac{1}{q_{-ii}} q_{-i} q_{-i}^\top \right) \mathbf{Y} = \left( Q^{-1} - \frac{1}{q_{-ii}} q_{-i} q_{-i}^\top \right) \mathbf{Y} \tag{45}$$

$$= \begin{bmatrix} R^{-1} \mathbf{Y}_{-i} \\ 0 \end{bmatrix}. \tag{46}$$

To show this, we plug in the inversion formula $Q^{-1}$ from equation 44 into equation 45 and using $q_{-ii} = \gamma^{-1}$:

$$\left( Q^{-1} - \frac{1}{q_{-ii}} q_{-i} q_{-i}^\top \right) \mathbf{Y} \tag{47}$$

$$= \left( \begin{bmatrix} R^{-1} + R^{-1} q_i q_i^\top R^{-1} & \gamma^{-1} R^{-1} q_i \\ \gamma^{-1} q_i^\top R^{-1} & \gamma^{-1} \end{bmatrix} - \frac{1}{\gamma^{-1}} \begin{bmatrix} \gamma^{-1} R^{-1} q_i \\ \gamma^{-1} \end{bmatrix} \begin{bmatrix} \gamma^{-1} R^{-1} q_i \\ \gamma^{-1} \end{bmatrix}^\top \right) \begin{bmatrix} \mathbf{Y}_{-i} \\ Y_i \end{bmatrix} \tag{48}$$

$$= \left( \begin{bmatrix} R^{-1} + \gamma^{-1} R^{-1} q_i q_i^\top R^{-1} & \gamma^{-1} R^{-1} q_i \\ \gamma^{-1} q_i^\top R^{-1} & \gamma^{-1} \end{bmatrix} - \gamma \begin{bmatrix} \gamma^{-2} R^{-1} q_i q_i^\top R^{-1} & \gamma^{-2} R^{-1} q_i \\ \gamma^{-2} q_i^\top R^{-1} & \gamma^{-2} \end{bmatrix} \right) \begin{bmatrix} \mathbf{Y}_{-i} \\ Y_i \end{bmatrix} \tag{49}$$

$$= \begin{bmatrix} R^{-1} + \gamma^{-1} R^{-1} q_i q_i^\top R^{-1} - \gamma^{-1} R^{-1} q_i q_i^\top R^{-1} & \gamma^{-1} R^{-1} q_i - \gamma^{-1} R^{-1} q_i \\ \gamma^{-1} q_i^\top R^{-1} - \gamma^{-1} q_i^\top R^{-1} & \gamma^{-1} - \gamma^{-1} \end{bmatrix} \begin{bmatrix} \mathbf{Y}_{-i} \\ Y_i \end{bmatrix} \tag{50}$$

$$= \begin{bmatrix} R^{-1} & \mathbf{0} \\ \mathbf{0}^\top & 0 \end{bmatrix} \begin{bmatrix} \mathbf{Y}_{-i} \\ Y_i \end{bmatrix} = \begin{bmatrix} R^{-1} \mathbf{Y}_{-i} \\ 0 \end{bmatrix}. \tag{51}$$

Therefore, we have equivalence between equation 42 and the formulation in the theorem.

$\square$

## A.5 An Analytical NTK Kernel

In this section, we provide theoretical results that allow deriving the closed form solution of the NTK for the two-layer ReLU network. First, see the paper by Jacot et al. Jacot et al. (2018) for the original derivation of the neural tangent kernel.

**A closed-form solution for two-layer ReLU networks.** From (Zhang & Zhang, 2021; Du et al., 2019, Assumption 3.1) we obtain the definition of the Kernel matrix $\mathbf{K}^\infty$ (termed Gram matrix in the paper Du et al. (2019)) for ReLU networks:

$$\mathbf{K}_{ij}^\infty = \mathbf{K}^\infty(\mathbf{x}_i, \mathbf{x}_j) = \mathbb{E}_{\mathbf{w} \sim \mathcal{N}(0, \mathbf{I})} \left[ \mathbf{x}_i^\top \mathbf{x}_j \mathbb{I} \left\{ \mathbf{w}^\top \mathbf{x}_i \geq 0, \mathbf{w}^\top \mathbf{x}_j \geq 0 \right\} \right]$$

$$= \mathbf{x}_i^\top \mathbf{x}_j \mathbb{E}_{\mathbf{w} \sim \mathcal{N}(0, \mathbf{I})} \left[ \mathbb{I} \left\{ \mathbf{w}^\top \mathbf{x}_i \geq 0, \mathbf{w}^\top \mathbf{x}_j \geq 0 \right\} \right]$$

$$= \mathbf{x}_i^\top \mathbf{x}_j \frac{\pi - \arccos \left( \frac{\mathbf{x}_i^\top \mathbf{x}_j}{\|\mathbf{x}_i\| \|\mathbf{x}_j\|} \right)}{2\pi}.$$

The last reformulation uses an analytical result by Cho & Saul (2009). The derived result matches the one by Xie et al. (2017), which however does not provide a comprehensive derivation.

## B  ADDITIONAL EXPERIMENTAL RESULTS

**Data sets for the Classification Tasks** When considering classification tasks on the *heloc* and *admission* data sets, we threshold the scores based on the median to obtain binary target labels. On the Admission data set (in the classification setting), a counterfactual is found when the predicted first-year average score switches from 'below median' to 'above median'. We then count an invalidation if, after the model update, the score of a counterfactual switches back to 'below median'. In addition to the aforementioned data sets, we use both the *Diabetes* and the *Compas* data sets. The *Diabetes* data set which contains information on diabetic patients from 130 different US hospitals (Strack et al., 2014). The patients are described using administrative (e.g., length of stay) and medical records (e.g., test results), and the prediction task is concerned with identifying whether a patient will be readmitted within the next 30 days. We sub sampled a smaller data sets of 10000 points from this dataset. 8000 points are left to train the model, while 2000 points are left for the test set. The *Compas* data set Angwin et al. (2016) contains data for more than 10,000 criminal defendants in Florida. It is used by the jurisdiction to score defendant's likelihood of reoffending. We kept a small part of the raw data as features like *name*, *id*, *casenumbers* or *date-time* were dropped. The classification task consists of classifying an instance into high risk of recidivism. Across all data sets, we dropped duplicate instances.

**Discussing the Results** As suggested in Section 6, here we are discussing the remaining recourse outcome invalidation results. We show these results for two settings. In Figure 4, we demonstrate the efficacy of our greedy deletion algorithm across 4 data sets on the classification tasks using different classification models (ANN, logistic regression, Kernel-SVM). For the logistic regression and the ANN model, we use the infinitesimal jackknife approximation to calculate the probitively expensive retraining step as described in Section 5. We observe that our method well outperforms random guessing. The results also highlight that while the NTK theory allows to study the deletion effects from a theoretical point of view, if one is interested in empirical worst-case approximations, the infinitesimal jackknife can be a method of choice. As we observe this pattern across all recourse methods, we hypothesize that this is related to the instability of the trained ANN models, and we leave an investigation of this interesting phenomenon for future work.

Additionally, in Figure 5, we compare our SGD-based deletion algorithm to the greedy algorithm. For the SGD-based deletion results, we observe inverse-u-shaped curves on some method-data-model combinations. The reason for this phenomenon can be explained as follows: when the $\ell_0$ regularization strength (i.e., $\eta$) is not strong enough, then the importance weights for the $k$-th removal with $k > 5$ become more variable (i.e., SGD does not always select the most important data weight for larger $k$). This drop in performance can be mitigated by increasing the strength of the $\ell_0$ regularizer within our SGD-based deletion algorithm.

In Figure 6, we study a simple removal strategy aimed at increasing the stability of algorithmic recourse. To this end, we identified the 15 points that lead to the highest invalidation on the NTK and linear regression models when the underlying recourse method is SCFE. Using our greedy method, we remove these 15 points from the training data set, and we then then rerun our proposed greedy removal algorithm. This strategy leads to an improvement of up to 6 percentage points over the initial model where the 15 most critical points were included, suggesting that the removal of these critical points can be used to alleviate the recourse instability issue. In future work, we plan to investigate strategies that increase the robustness of algorithmic recourse even further.

Finally, in Figure 7 we study how well the critical points identified for the NTK model would invalidate a wide 2-layer ReLU network with 10000 hidden nodes. To study that question, we identified the points that lead to the highest invalidation on the NTK using our greedy method, and we then use these identified training points to invalidate the recourses suggested by the wide ANN. As before, we are running these experiments on the full data set across 5 folds. Figure 7 demonstrates

the results of this strategy for the `SCFE` recourse method. We see that this strategy increases the robustness of up to 30 percentage points over the random baseline, suggesting that critical points under NTK can be used to estimate recourse invalidation for wide ANN models.

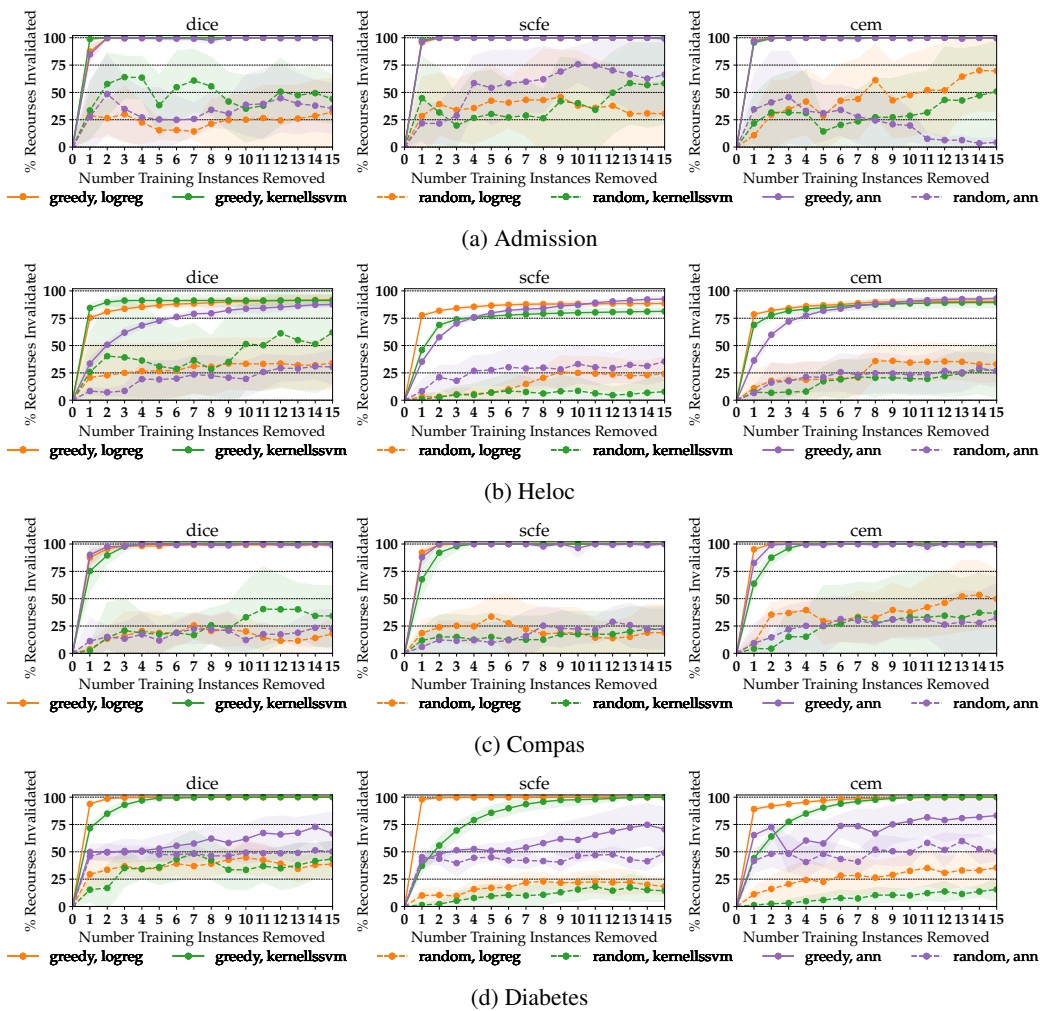

Figure 4: Measuring the tradeoff between recourse outcome instability and the number of deletion requests for the Admission, Heloc, Diabetes and Compas data sets for logistic regression, kernel svm, and ANN models across recourse methods on classification tasks. Results were obtained by greedy optimization. The dotted lines indicate the random baselines.

## C  IMPLEMENTATION DETAILS

### C.1  DETAILS ON MODEL TRAINING

We train the classification models using the hyperparameters given in Table 1. The ANN and the Logistic regression models are fit using the quasi-newton `lbgfs` solver. We add L2-regularization to the ANN weights. The other methods are trained via their analytical solutions. Below, in Algorithms 1 and 2, we show pseudocodes for both our greedy and sgd-based deletion methods to invalidate the recourse outcome. In order to do the optimization with respect to the recourse action stability measure, we slightly adjust Algorithm 2 to optimize the right metric from Definition 2.

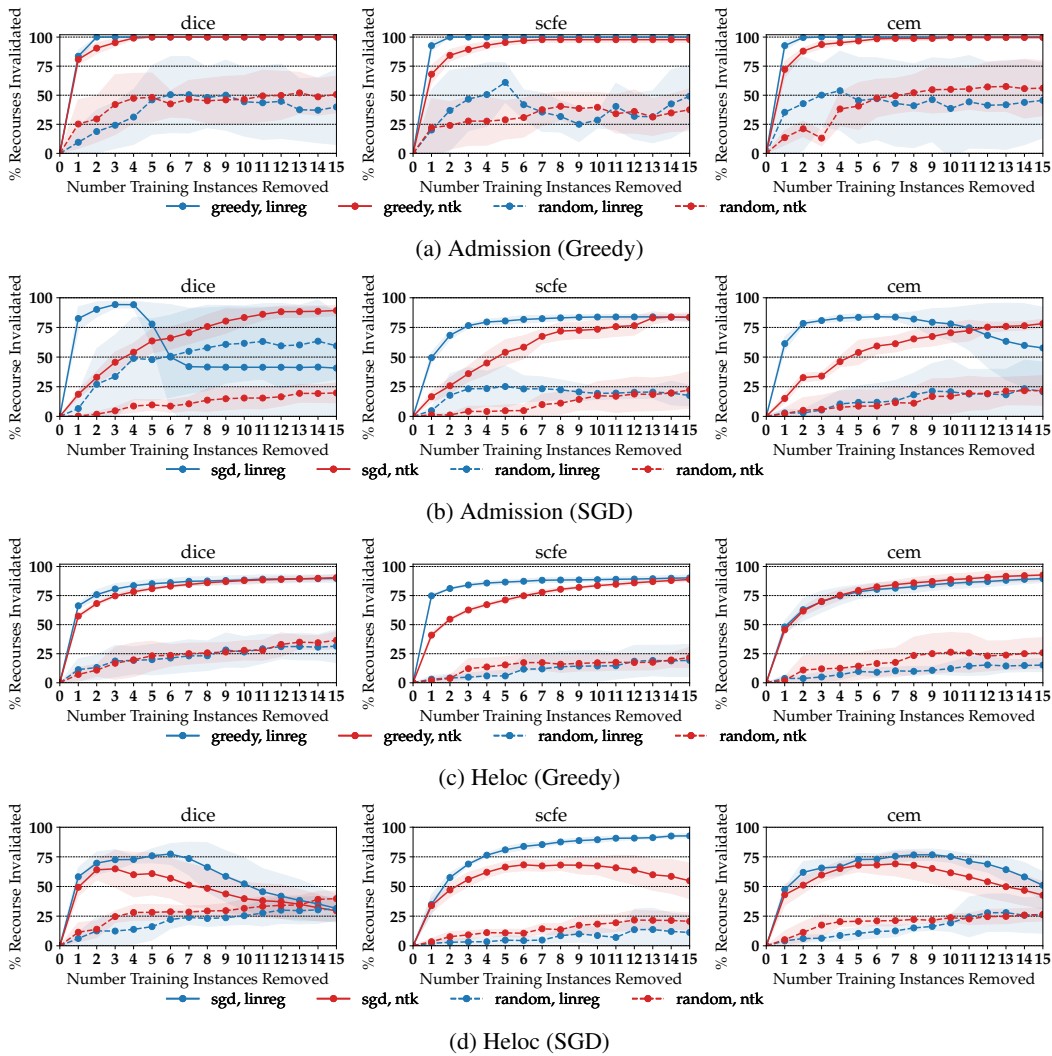

Figure 5: Measuring the tradeoff between recourse outcome instability and the number of deletion requests for the Admission and Heloc data sets for linear regression and NTK models across recourse methods on regression tasks. Results were obtained by both SGD and Greedy optimization. The dotted lines indicate the random baselines.

## C.2 DETAILS ON GENERATING THE COUNTERFACTUALS

For DICE, for every test input, we generate two different counterfactual explanations. Then we randomly pick either the first or second counterfactual to be the counterfactual assigned to the given input. Across all recourse methods the success rates lie above 95%, i.e., for 95% of recourse seeking individuals the algorithms can identify recourses. The only exception is admission data set for the NTK model, where the success rate lies at 60%. Across all recourse methods we set $\lambda \to 0$. Note that the default implementations use early stopping once a feasible recourse has been identified.

## C.3 DETAILS ON THE $\ell_0$ REGULARIZER

Since an $\ell_0$ regularizer is computationally intractable for high-dimensional optimization problems, we have to resort to approximations. One such approximation approach was recently suggested by Yamada et al. (2020). The underlying idea consists of converting the combinatorial search problem to a continuous search problem over distribution parameters. To this end, recall our optimization

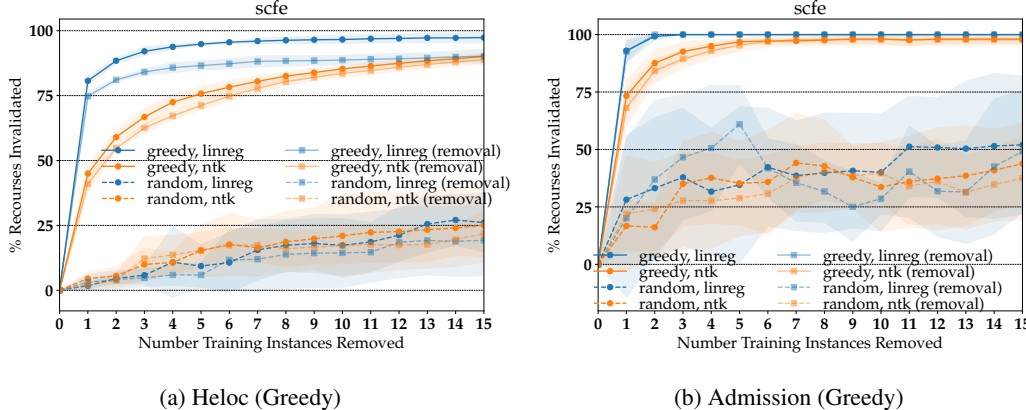

(a) Heloc (Greedy)  (b) Admission (Greedy)

Figure 6: Measuring the efficacy of a simple removal strategy on the Heloc and Admission data set for linear and NTK regression models. We removed the 15 critical points identified for the linear and NTK models when the underlying recourse method is SCFE and reran the removal algorithm on the remaining training set. Results were obtained by Greedy optimization. The dotted lines indicate the random baselines.

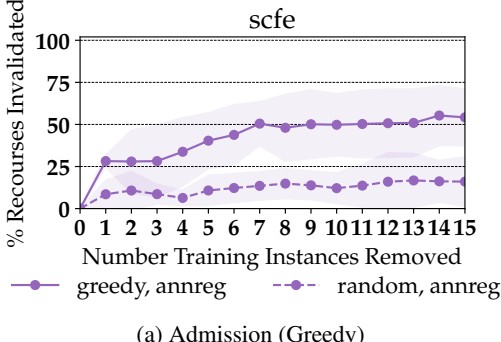

(a) Admission (Greedy)

Figure 7: Measuring the tradeoff between recourse outcome instability and the number of deletion requests for the Admission data set for a neural network regression model. We used the critical points identified for the NTK model to invalidate the recourses identified by a wide 2-layer ReLU network with 10000 hidden nodes. Results were obtained by Greedy optimization. The dotted lines indicate the random baselines.

problem from the main text:

$$\boldsymbol{\omega}^* = \underset{\boldsymbol{\omega} \in \{0,1\}^n}{\arg \max} \; m(\boldsymbol{\omega}) - \eta \cdot \|\mathbf{1} - \boldsymbol{\omega}\|_0. \tag{52}$$

We will now introduce Bernoulli random variables $Z_i \in \{0, 1\}$ with corresponding parameters $\pi_i$ to model the individual $\omega_i$. Instead of optimizing the objective above with respect to $\boldsymbol{\omega}$ we will optimize

| Model | Parameters |
|---|---|
| Linear Regression | OLS, no hyperparameters. |
| NTK Regression | $\beta = 2$ (Admission), $\beta = 5$ (other data sets) |
| Logistic Regression | L2-Regularization with $C = 1.0$ |
| Kernel-LSSVM | Gaussian Kernel with $\gamma = 1.0$ (see Cawley & Talbot (2004)) |
| ANN | 2-Layer, 30 Hidden units, Sigmoid, $\alpha = 10$ (L2-Regularization) |

Table 1: Model hyperparameters used in this work

---

**Algorithm 1** Greedy recourse outcome invalidation

---

**Required:** Model: $f_{\mathbf{w(1)}}$; Matrix of Recourses: $\check{\mathbf{X}}_\mathbf{1} \in \mathbb{R}^{q \times d}$; $d$: input dimension; $q$ number of recourse points on test set; $n$: # train points; $M$: max # deleted train points; $s$: invalidation target

$\boldsymbol{\omega}^{(0)} = \mathbf{1}_n$             $\triangleright$ All training instances present

**for** $m = 1 : M$ **do**

    $\boldsymbol{\omega}^{(m)} \leftarrow \boldsymbol{\omega}^{(m-1)}$

    $\tilde{\mathbf{Y}} = \mathbf{0}_{n \times q}$                $\triangleright$ Recourse outcomes

    $\boldsymbol{J} = \mathbf{0}_{n \times q}$                $\triangleright$ Invalidations present

    $S^{(m)} \leftarrow \left\{ i \;\middle|\; \boldsymbol{\omega}_i^{(m)} \neq 0 \right\}$       $\triangleright$ Set of train instances present at iteration $m$

    **for** $i \in S^{(m)}$ **do**

        $\mathbf{w}_{\text{new}}^{(i)} = \texttt{update\_w}(\boldsymbol{\omega}_{-i}^{(m)})$     $\triangleright$ $\boldsymbol{\omega}_{-i}^{(m)}$ has additionally set weight $i$ to 0.

                                       $\triangleright$ Use analytical or IJ solution for $\mathbf{w}(\boldsymbol{\omega})$

        $\tilde{\mathbf{Y}}[i,:] = f_{\mathbf{w}_{\text{new}}^{(i)}}(\check{\mathbf{X}}_\mathbf{1})$             $\triangleright$ New recourse outcomes

        $\boldsymbol{J}[i,:] = \mathbb{I}(\tilde{\mathbf{Y}}[i,:] < s)$          $\triangleright$ Invalidation present

    **end for**

    index $\leftarrow \; \arg\max_{i \in S^{(m)}} \|\mathbf{J}[i,:]\|_1$    $\triangleright$ Find point that leads to highest invalidation

    $\boldsymbol{\omega}^{(m)}[\text{index}] = 0$              $\triangleright$ Remove training point

**end for**

**return**: $\boldsymbol{\omega}^{(M)}$             $\triangleright$ data weights indicating $M$ removals

---

with respect to distribution parameters $\boldsymbol{\pi}$:

$$\boldsymbol{\pi}^* = \arg\max_{\boldsymbol{\pi}} \; m(\mathbf{Z}(\boldsymbol{\pi})) - \eta \cdot \|\mathbf{1} - \mathbf{Z}(\boldsymbol{\pi})\|_0. \tag{53}$$

Since the above optimization problem is known to suffer from high-variance solutions, (Yamada et al., 2020) suggest to use a Gaussian-based continuous relaxation of the Bernoulli variables:

$$\tilde{Z}_i = \max(0, \min(1, \mu_i + \epsilon_i)), \tag{54}$$

where $\epsilon_i = \mathcal{N}(0, \sigma^2)$, resulting in the following optimization problem:

$$\boldsymbol{\mu}^* = \arg\max_{\boldsymbol{\mu}} \; m(\tilde{\mathbf{Z}}(\boldsymbol{\mu})) - \eta \cdot \|\mathbf{1} - \tilde{\mathbf{Z}}(\boldsymbol{\mu})\|_0. \tag{55}$$

At inference time, the optimal weights are then given by $\tilde{Z}_i^* = \max(0, \min(1, \mu_i^*)) \; \forall i \in [n]$. To obtain discrete weights, we take the argmax over each individual $\tilde{Z}_i$.

### C.4    DETAILS ON THE JACKKNIFE APPROXIMATION

When the model parameters $\mathbf{w}$ are a function of the data weights by solving equation 1 we can approximate $\mathbf{w}(\boldsymbol{\omega})$ using the infinitesimal Jackknife (IJ) without having to optimize equation 1 repeatedly (Jaeckel, 1972; Efron, 1982; Giordano et al., 2019b;a):

$$\mathbf{w}_{\text{IJ}}(\boldsymbol{\omega}) = \mathbf{w}_\mathbf{1} - \mathbf{H}_\mathbf{1}^{-1} \mathbf{G}_{\boldsymbol{\omega}-1}, \tag{56}$$

where $\mathbf{G}$ and $\mathbf{H}_\mathbf{1}$ are the Jacobian and the Hessian matrices of the loss function with respect to the data weights evaluated at the optimal model parameters $\mathbf{w}$, i.e., $\mathbf{G}_{\boldsymbol{\omega}-1} = \frac{1}{n} \sum_{i=1}^{n} (\omega_i - 1) \cdot \frac{\partial \ell(f_\mathbf{w}(\mathbf{x}_i), y_i)}{\partial \mathbf{w}}$ and $\mathbf{H}_\mathbf{1} = \frac{1}{n} \sum_{i=1}^{n} \frac{\partial^2 \ell(f_\mathbf{w}(\mathbf{x}_i), y_i)}{\partial \mathbf{w} \partial \mathbf{w}^\top}$. Note that this technique computes the Hessian matrix $\mathbf{H}_\mathbf{1}$ only once. Using this Jackknife approximation, the Jacobian term $\mathbf{G}_{\boldsymbol{\omega}-1}$ becomes an explicit function of the data weights which makes the Jackknife approximation amenable to optimization.

---

**Algorithm 2** SGD recourse outcome invalidation

---

**Required:** Model: $f_{\mathbf{w(1)}}$; Matrix of Recourses: $\check{\mathbf{X}}_\mathbf{1} \in \mathbb{R}^{q \times d}$; $d$: input dimension; $q$ number of recourse points on test set; $n$: # train points; $M$: max # deleted train points; $s$: invalidation target

$\boldsymbol{\mu}^{(1)} = \mathbf{1}_n$      ▷ Mu are soft data weights that are opimized.

**for** $m = 1 :$ Step **do**      ▷ Perform $Step$ number of updates.

     $\delta-$loss=0.0

     **for** $k = 1 : K$ **do**      ▷ Use $K$ Monte-Carlo Samples for the approximation

         Sample $\boldsymbol{\epsilon}_k^{(m)} \sim \mathcal{N}\left(0, \sigma^2 \mathbf{I}_n\right)$

         $\boldsymbol{\omega}_k^{(m)} = \max\left(0, \min\left(1, \boldsymbol{\mu}^{(m)} + \boldsymbol{\epsilon}_k^{(m)}\right)\right)$ ▷ Sample (soft) data weights as in Yamada et al. (2020)

         $\mathbf{w}_{k,\text{new}}^{(m)} = \texttt{update\_w}(\boldsymbol{\omega}_k^{(m)})$      ▷ Compute model weights from data weights either analytically or with IJ

         $l_k^{(m)} = \text{sigmoid}\left(f_{\mathbf{w}_{k,\text{new}}^{(m)}}\left(\check{\mathbf{X}}_\mathbf{1}\right) - s\right)$      ▷ Predict with new weights and compute soft invalidation.

         $\delta-$loss $= \delta-$loss $+ \|l_k^{(m)}\|_1$      ▷ Add up soft inval. loss

     **end for**

     $r^{(m)} = \sum_{i=1}^n \Phi\left(\frac{1-(\boldsymbol{\mu}^{(m)})_i}{\sigma}\right)$      ▷ Sparsity Regularizer from Yamada et al. (2020)

     $\boldsymbol{\mu}^{(m+1)} = \boldsymbol{\mu}^{(m)} + \gamma \nabla_{\boldsymbol{\mu}^{(m)}}\left(\frac{\delta-\text{loss}}{D} + \lambda r^{(m)}\right)$      ▷ Grad. Descent with lr. $\gamma$

**end for**

removed_ind $= \texttt{argsort}(\boldsymbol{\mu}^{(\text{Step}+1)})$      ▷ Sort indices ascendingly

$j = 0$

$\boldsymbol{\omega} = \mathbf{1}_n$

**while** $j < M$ **do**      ▷ Binarize and fulfil max number M

     $\boldsymbol{\omega}$[removed_ind[j]]$= 0$

     $j = j + 1$

**end while**

**return**: $\boldsymbol{\omega}$      ▷ data weights indicating $M$ removals

---

