# OpenReview forum: "On the Trade-Off between Actionable Explanations and the Right to be Forgotten"
_ICLR.cc/2023/Conference — ICLR 2023 poster_

### Official Review · Reviewer_CkVr · 2022-10-17

**Confidence:** 3
**Correctness:** 4
**Technical Novelty And Significance:** 4
**Empirical Novelty And Significance:** 3
**Recommendation:** 6

**Clarity, Quality, Novelty And Reproducibility:**

### Clarity and Quality:

The paper is generally clearly and well written.

### Novelty:
To the best of my knowledge, the consideration of the trade-off between algorithmic resource and the right to be forgotten and the recource instability measures are novel.

### Reproducibility:
Reproducibility should not be an issue as the authors provide sufficient details of the experiments and the code as well. I quickly scanned through the code but did not run it.

**Strength And Weaknesses:**

### Strengths

- The studied problem is novel and interesting. Both algorithmic resource and the right to forgotten are very relevant and important.
- The perspective of considering resource instability is interesting.
- The paper is generally well written and organized.

### Weaknesses
- The connection between the setting of ERM in Section 3 and the NTKs in Section 4 can be made more explicit. While I understand that NTKs are probably adopted here because they offer the theoretical result equation (7) and the subsequent results, perhaps a simpler learning setting where the model parameters (and thus updates) are more analytically tractable and can explicitly leverage the ERM setting would make the argument more grounded. For instance, there are a few places the authors used the concepts of influence, leverage. I wonder whether following the simpler setting (i.e., linear regression) to leverage these well-understood concepts would enable a more in-depth and "tighter" analysis.


- The key result in Section 4 depends on neural tangent kernel, which may give a looser theoretical result than ideal. Specifically, there are a few steps of approximations necessary for NTKs (e.g., $k \to \infty$). While this may be a weaknesss of NTK itself, highlighting this point or in what way the approximations in NTK can limit the analysis. For instance, the authors mention
> we provide data dependent upper bounds on the invalidation measures from Definitions 1 and 2, which practitioners can use to probe the worst-case vulnerability of their algorithmic recourse to data deletion requests

    It would be nice to help the readers gauge the tightness of this worst-case vulnerability. Right now, while the upper bounds are a nice theoretical result, how they should be applied in practice can be explained further (e.g., are the upperbounds likely to be overly pessimistic, what are the situations these might not hold).

- Section 5 is proposed to specifically tackle the computational challenges arising from (i) possible retraining from deletion; (ii) a combinatorial subset selection to maximize the instability. It would be more reassuring that some theoretical guarantees are provided, especially if the dataset is large and/or the model is complex (as motivated earlier where the number of hidden node $k\to \infty$).

**Summary Of The Paper:**

The paper studies the trade-off between the ability to achieve algorithmic resource vs. the right to be forgotten in machine learning, specifically via linear models and wide neural networks. It illustrates that these two desiderate (algorithimic resource and the right to be forgotten) are fundamentally at odds by defining two measures of recourse instability. The paper further devises an algorithm to demonstrate this by opimizting for the minimum number of data points to forget that result in the highest instability. Empirical results on two real-world datasets are provided.

**Summary Of The Review:**

I think the paper takes an interesting approach to a relevant problem. Though the choice of the particular theoretical/technical framework can be improved to make the argument more grounded, I believe there are sufficient contributions both theoretical and empirical. The paper is generally well written with clarity.

---

> ### Author Response · Authors · 2022-11-11
> **Response to Reviewer CkVr**
>
> Thank you for the thoughtful comments and suggestions and for seeing the relevance of our work.
>
> > I wonder whether following the simpler setting (i.e., linear regression) to leverage these well-understood concepts would enable a more in-depth and "tighter" analysis.
>
> Thank you for raising this suggestion. We follow this train of thought in the beginning of Section 4 (Proposition 1 and Corollary 1), where we explicitly study the linear model obtained through empirical risk minimization (ordinary least squares). The results obtained with the NTK in the subsequent Proposition 2 follow a similar intuition (this time in the feature space defined by the kernel): instances which have influential labels, are atypical, and cannot be well fit by the model will have the highest impact.
>
>
> > It would be nice to help the readers gauge the tightness of this worst-case vulnerability. Right now, while the upper bounds are a nice theoretical result, how they should be applied in practice can be explained further (e.g., are the upper bounds likely to be overly pessimistic, what are the situations these might not hold)
>
> Considering a random removal strategy, the upper bounds will likely be quite pessimistic. Considering our SGD or greedy based adversarial removal strategies, our upper bounds are expected to be relatively tight, as we have identified two necessary conditions for tightness:
> - The recourse action and the change of the decision boundary show in the same direction. For the NTK, this condition changes slightly to: The recourse action in feature space and the change of the decision boundary show in the same direction.
> - We exactly remove the point with maximal influence.
>
> In this – not unrealistic – case, our bound will be an exact approximation of the recourse outcome instability for linear and NTK models.
>
> > Section 5 is proposed to specifically tackle the computational challenges arising from (i) possible retraining from deletion; (ii) a combinatorial subset selection to maximize the instability. It would be more reassuring that some theoretical guarantees are provided, especially if the dataset is large and/or the model is complex
>
> We would like to highlight that the problem stated in equation (14) is NP-hard in the general case (Appendix A.3), which is why we cannot give guarantees on the computational complexity for the original problem stated in equation (14). However, we can investigate the complexity of the approximation algorithm in Section 5. Our greedy approximation algorithm requires $O(\alpha n^2)$ model weight computations / approximations where $\alpha$ controls the fraction of instances that are being removed from the training set. The final complexity is dependent on that of the weight recomputation step.

---

> > ### Comment · Reviewer_CkVr · 2022-11-17
> > **Post rebuttal**
> >
> > I thank the authors for the clarifications. My questions are addressed.
> >
> > One comment on writing w.r.t. linear models. It might be a bit confusing to put the linear models and neural networks both under the analysis using NTKs. Correct me if I am wrong, Proposition 1 (w.r.t. linear models) does not really need NTKs at all. Moreover, to use NTKs, it seems one needs the ReLU activation function as described in equation (5), but often the linear model for classification is logistic regression, which does not typically use ReLU activation function. Therefore, it may help improve the clarity is this distinction is clearly highlighted to the user. For both models, a common intuition is applied but the technical tools to make this intuition precise (for both models) are not exactly the same.

---

> > > ### Author Response · Authors · 2022-11-18
> > > **Follow up response**
> > >
> > > Thank you for your constructive feedback. Following your suggestion, in the updated version we have now presented the theoretical results on linear models first before we introduce the NTK and the corresponding robustness guarantees.
> > >
> > > We were glad to learn that we could appropriately address your questions and that you acknowledged the correctness and technical novelty of our work with the highest scores. In light of these clarifications, we would highly appreciate if the reviewer would reconsider their overall score of the paper.

---

### Official Review · Reviewer_2k5p · 2022-10-20

**Confidence:** 3
**Correctness:** 4
**Technical Novelty And Significance:** 3
**Empirical Novelty And Significance:** 3
**Recommendation:** 6

**Clarity, Quality, Novelty And Reproducibility:**

Clarity: pretty good, problem is clearly motivated. Some minor claims are not totally clear to me (see previous section of review)
Quality: good, the paper is neatly written and the results and experiments seem correct, albeit not mind-blowing
Novelty: definitely a novel problem formulation
Reproducibility: I think this would probably be reproducible, information is given on hyperparameters in the paper and the code is provided

**Strength And Weaknesses:**

Strengths:
- a novel direction, combines two questions which I have not seen combined before (recourse + deletion)
- Beyond responsible AI-type questions, this research direction could have more general ramifications for understanding the shape of the decision-boundary in e.g. an NTK
- the theoretical and experimental work both provide clear, concrete contributions to the central question of the paper

Weaknesses:
- I'm not sure the theoretical work provides that much insight - it seems that the main takeaway is that points with higher influence are going to provide more recourse instability, which seems obvious. I'm not saying they're bad results (it's good to have them) but not sure they are that revealing. Maybe there is intuition here which can be expanded on?
- would be good to draw more comparisons to related work looking at robustness of explanations/recourse, since that's what this paper is essentially about
- I'd like to know more about what happens in the average case of deletion, since in practice that's what we'd mostly be looking at (rather than the worst-case)
- the logic on page 6 below Corollary 1 confuses me: why would we assume that the parameter norm's change will be minimal, but not assume that the actual difference change will be minimal? I understand the two are not equivalent, I'm just not sure why the authors choose to make one assumption and not the other
- the algorithms in 5.2 seem very computationally intensive, requiring the re-learning of the model after every point removal - I'm wondering if the authors could expound more on these challenges practically, or discuss if they think there are ways around this
- "Evaluation Measures" paragraph refers to the gradient-based method but I don't see it in Fig 2
- "... by up to 6 percentage points" - I'm not sure I understand this result, might need more clarification in the paper
- on page 9 (in conclusion, and possibly in the last paragraph of Sec 6 as well), an approach is discussed where you remove the highest influence points from training. I don't totally understand how you could do this before the model is trained - is that the intention?

**Summary Of The Paper:**

In this paper, the authors discuss the properties of recourse-type counterfactual explanations under model perturbation by removing a data point. Drawing a line to the impact this would have on the usefulness of recourse recommendations in a regime where data deletion occurs post-training, they specify two ways in which explanations can be non-robust under this perturbation - the usefulness of an old explanation under a new (post-deletion) model, and the change in an explanation pre- and post-deletion. They upper bound these impacts in the case of a linear model and an NTK, and provide heuristic approaches for finding the set of points whose deletion would cause the most instability in explanations and their outcomes. They provide some experimental validation, showing in particular that you can find very small subsets which "invalidate" recourses - that is, which cause old recourses to not provide the desired outcome under the new model.

Note: I think I reviewed this paper in a past conference, so apologies if some of this content is familiar. I did not go back and look at that review before re-reading the paper, so I had fresh eyes on it, although its plausible some of the feedback is similar.



**Summary Of The Review:**

I think the novelty of this setup combined with the clarity and quality of the writeup and basic results are sufficient to recommend acceptance. I don't necessarily feel the paper goes the "extra mile" to be a slam dunk accept - the theoretical and experimental results are mostly as I would have expected and I don't necessarily feel like they give me tons of extra insight. However this does not make them bad results and it's good to have them done.

---

> ### Author Response · Authors · 2022-11-11
> **Response to Reviewer 2k5p**
>
> Thank you for the time spent reviewing our work. We appreciate the thoughtful comments and suggestions.
>
> >(1) [...] it seems that the main takeaway is that points with higher influence are going to provide more recourse instability, which seems obvious. [...] Maybe there is intuition here which can be expanded on?
>
> >(2) [...] would be good to draw more comparisons to related work looking at robustness of explanations/recourse, since that's what this paper is essentially about
>
> The robustness of algorithmic recourse depends on the robustness of the predictive model to the deletion of a specific instance (influence). However, our work is the first to establish this fundamental link between data point influence and the robustness of counterfactual explanations. Moreover, it explicitly quantifies this link through theoretical results. This has not been demonstrated before, and has real-world implications:
>
> - Our theory also suggests a fundamental paradigm shift in the way that the recourse literature thinks about robustness of algorithmic recourse: most of the literature follows the paradigm of developing robust recourse methods taking the predictive model as fixed [1-4]. This usually leads to recourses that are more difficult to act upon as they are less parsimonious and have higher recourse costs at the benefit of higher robustness [1-4]. In contrast to this paradigm, our analysis suggests that one can achieve an increased levels of robust recourse by training the model in such a way that has lower influence functions.
>
> - Thus, our theory can be operationalized with some approaches to minimize influence but which have been proposed in a different context, for instance private or deletion-robust models. Our work builds the bridge between these two fields, which we see as a valuable contribution.
>
> > I'd like to know more about what happens in the average case of deletion, since in practice that's what we'd mostly be looking at (rather than the worst-case)
>
> Note that our experiments cover the average case as well (i.e., the random baselines in Figures 2 and 3 correspond to the “average case”). They show that removing up to two points can lead to considerable invalidations of close to 25 percent (see Figure 2).
>
> >The logic on page 6 below Corollary 1 confuses me. Why would we assume that the parameter norm's change will be minimal, but not assume that the actual difference change will be minimal?
>
> In principle, we do not need to make any further simplification after corollary 1. After corollary 1 we make this  simplification to merely ease the interpretation of corollary 1. Usually, parameter vectors of a high dimensional space will have large norms. Therefore, the change, $||d_i||$ can be significantly $>>0$ but negligible in terms of the total norm $||w_{-i}||$. If we were to assume $||d_i|| = 0$, this would mean no model changes are possible. This is unrealistic and would result in no change in recourse. Therefore these two ways of further simplification are non-equivalent and lead to fundamentally different results.
>
>
> >[...] the algorithms in 5.2 seem very computationally intensive, requiring the re-learning of the model after every point removal [...]
>
> For linear and NTK models retraining is feasible via analytical solutions, and no extensive optimization is required.
>
> Using the Infinitesimal Jackknife approximation instead of full model retraining allows us to identify an entire set of critical points without the re-learning of the model after every removal: For complex models such as ANNs the most computationally demanding component is the re-calculation of the model parameters depending on a set of data weights. Instead of performing full extensive optimization one can resort to the Infinitesimal Jackknife approximation. This approximation requires no model retraining compared to the greedy algorithm since it is derived from a model trained on the full data set. However, as we move further away from the point of expansion ($\omega = \mathbf{1}$) the approximation quality deteriorates, and retraining the model might still be necessary after several SGD / greedy  steps.
>
>
>
> > Evaluation Measures" paragraph refers to the gradient-based method but I don't see it in Fig 2
>
> SGD based results are shown in Appendix B and Figure 3.
>
> ----
> [1] E. Black, Z. Wang, M. Fredrikson, and A. Datta. Consistent counterfactuals for deep models. arXiv:2110.03109,, 2021
>
> [2] S. Upadhyay, S. Joshi, and H. Lakkaraju. Towards robust and reliable algorithmic recourse, NeurIPS , volume 34, 2021
>
> [3] M. Pawelczyk, T. Datta, J. van-den Heuvel, G. Kasneci, and H. Lakkaraju. Algorithmic recourse in the face of noisy human responses. arXiv preprint arXiv:2203.06768 , 2022
>
> [4] R. Dominguez-Olmedo, A.-H. Karimi, and B. Schölkopf. On the adversarial robustness of causal algorithmic recourse, ICML, 2022.

---

> > ### Author Response · Authors · 2022-11-11
> > **Response to Reviewer 2k5p continued**
> >
> > >[...] an approach is discussed where you remove the highest influence points from training. I don't totally understand how you could do this before the model is trained - is that the intention?
> >
> > The intention of this method is to analyze a trained candidate model for non-robust data samples. After identification of instability with the methods proposed in this work, one can train a stable model without these points. We observed that this leads to an increase in robustness. To be more precise, we identified the 15 points that lead to the highest invalidation of recourse. Using our greedy method, we remove these 15 points from the training data set, and we then rerun our proposed greedy removal algorithm. This strategy leads to an improvement of up to 6 percentage points over the initial model where the 15 most critical points were included, suggesting that the removal of these critical points can be used to alleviate the recourse instability issue.

---

> > ### Comment · Reviewer_2k5p · 2022-11-17
> > **Response**
> >
> > Thanks for the rebuttal - I appreciate the clarifications. The point about the norm vs difference change is clarified for me, would be worthwhile making that just a little more explicit in the text.
> >
> > The new experiment is interesting! This is probably worth highlighting a bit more in the text if you're able to, it lends more credence to the method's applicability I think.

---

### Official Review · Reviewer_8JMg · 2022-10-23

**Confidence:** 4
**Correctness:** 3
**Technical Novelty And Significance:** 3
**Empirical Novelty And Significance:** 3
**Recommendation:** 6

**Clarity, Quality, Novelty And Reproducibility:**

The paper’s focus is on the theoretical understanding of the trade-off between the right to be forgotten and recourse invalidation. It is nice to have the recourse instability notions formalized first, and analyzed later for worst-case performance (vulnerability).

Below are some concerns I have:

1. I feel connections should be drawn between the accuracy and explanation aspects of deletion. When there exist works that delete data without compromising model accuracy, in their context, deletion robustness aims to prevent significant change in model parameters (i.e., no large $d_i$, the influence). It is not so different here when we consider recourse outcomes that are also linked to model parameter changes. Hence, the novelty of this work is lowered as the explanation framework is another deletion-robust problem in disguise. Could you explain why is deletion robustness insufficient?

2. It is nice to have comparisons for the robustness of different model classes in the experiments. It would be nice to see another application that the authors mention, which is the use of the method to perform data minimization for a model more robust to deletion. For example, is the model trained with the most influential points removed more robust now? Or, would it suggest the exclusion of certain input features, which is also an aspect of data minimization?

3. The definition of $\Delta$ and $\Phi$ in Definition. 1 and 2 are general to data weights $\omega$. However, the $\Delta$ and $\Phi$ in the Propositions refers to instability with respect to a single instance deletion. This may cause confusion and consider changing. Also, can the Propositions be generalized to multiple instances deletion?

4. For proposition 2, wouldn’t the NTK matrix $K^\infty(X,X)$ change in dimension when a data instance is removed? Would this cause a problem to the derivation in Equation (21) in Appendix A.1?

5. It is understandable and reasonable to conduct theoretical analysis on the simplest model classes as an initial investigation. Note that both “the right to be forgotten” and “explainability” stems from practical problems, it is necessary to at least empirically study the larger and more complex neural networks we use in practice? Are they much less susceptible to removal? Are the recourses then less vulnerable to deletion requests?

**Strength And Weaknesses:**

Strengths:
1. The paper calls the newly formulated research problem that investigates connections between explainability and the right to be forgotten.
2. The paper is well-written, clear and presented with interpretable results throughout.
3. The paper presents several theoretical and practical insights towards building deletion robust applications from the perspective of both data and model.

Strengths:
1. The paper calls the newly formulated research problem that investigates connections between explainability and the right to be forgotten.
2. The paper is well-written, clear and presented with interpretable results throughout.
3. The paper presents several theoretical and practical insights towards building deletion robust applications from the perspective of both data and model.

**Summary Of The Paper:**

This paper presents an initial study on the trade-off and connections between two key data regularization principles, “the right to be forgotten” and “the right to recourse”. The authors formalize the new recourse robustness problem through outcome and action instability, and subsequently upper bounds the recourse instabilities caused by a single data instance removal on simple (kernelized) linear models. The paper also proposed algorithms to efficiently find the critical data points responsible for the instabilities. Empirical experiments show that simple models considered in this paper are very susceptible to data deletion requests and not stable in terms of recourse actions. Also, the algorithm proposed is effective in finding those critical points.

**Summary Of The Review:**

Overall, I like the refreshing idea that relates data deletion to actionable explanations, while efforts are still needed to better distinguish this problem from other aspects already studied under data deletion (e.g., accuracy). I also think the theoretical analysis sheds insights into the direct link between data point influence and counterfactual explanation instability. The clear writing is also a bonus. Some points mentioned above can still be improved.

---

> ### Author Response · Authors · 2022-11-11
> **Response to Reviewer 8JMg**
>
> We thank the reviewer for their detailed feedback and are glad to see that the reviewer appreciates the "refreshing idea" investigated in this work.
>
> > It would be nice to see another application that the authors mention, which is the use of the method to perform data minimization for a model more robust to deletion. For example, is the model trained with the most influential points removed more robust now?
>
> We have investigated the removal of the most influential points on the Heloc and the Admission data sets and the corresponding effect on the counterfactual robustness in Appendix B (see Figure 6).  In particular, we identified the 15 points that lead to the highest invalidation of recourse. Using our greedy method, we remove these 15 points from the training data set, and we then rerun our proposed greedy removal algorithm. This strategy leads to an improvement of up to 6 percentage points over the initial model where the 15 most critical points were included, suggesting that the removal of these critical points can be used to alleviate some of the recourse instability issues.
>
>
> > [...] can the Propositions be generalized to multiple instances deletion?
>
> Currently, in our theory we consider only a single weight, however, our approach can be easily extended to removals of a subset of data points. In particular, one has to substitute $w - w_{-i}$ in the proof of Proposition 1 by $w - w_{-\mathcal{S}}$, where $\mathcal{S}$ denotes a set of $k$ training data indices to be removed from the training set. For example, for linear regression models we want to identify $w - w_{-S}$. To do this, define $X_{-\mathcal{S}}$ as the matrix of containing $k$ input points that shall be removed from the optimal weight vector $w=(X^\top X)^{-1} X^\top Y$, and let $Y_{-\mathcal{S}}$ be their corresponding labels. Then, the leave-k-out estimator is given by $w - w_{-\mathcal{S}} = (X^T X)^{-1} X_{\mathcal{S}}^T (I_k-H_{\mathcal{S}})^{-1} (Y_\mathcal{S} - X_\mathcal{S} w)$, where $H_{\mathcal{S}} = X_{\mathcal{S}}^T (X^T X)^{-1} X_{\mathcal{S}}$. A similar derivation can be worked out for the NTK. We have refrained from this approach because, compared to the case where only one data point is being deleted, the theoretical statement becomes more difficult to interpret while allowing for the same takeaways as in the signal deletion case.
>
>
> > For proposition 2, wouldn’t the NTK matrix K∞(X,X) change in dimension when a data instance is removed? Would this cause a problem to the derivation in Equation (21) in Appendix A.1?
>
> At inference time, the full data NTK model can be written as $K(x, X)^Tw_{ntk}$, where $w_{\text{ntk}}$ is of length $n$ and where $w_{\text{NTK}} = \big(K^\infty(X, X)  + \beta I_n \big)^{-1} Y$. As you suggested, the model with a deleted instance can as well be expressed by a weight vector of length $n-1$. However, as you correctly realized, your suggested approach would be impractical to compute the parameter update with respect to the previous model’s weight vector. To ensure that $w_{\text{ntk}}$ has fixed length $n$, the updated $w_{\text{ntk}}$ in equation (21) automatically features some changes to the weights corresponding to the remaining data instances, while the weight corresponding to the deleted data instance will automatically be exactly zero. This allows us to compute the weight change while remaining in the $n$ dimensional space. Please let us know if you have further questions regarding this aspect
>
>
> > Note that both “the right to be forgotten” and “explainability” stems from practical problems, it is necessary to at least empirically study the larger and more complex neural networks we use in practice?
>
> Yes, the reviewer is absolutely right. This is why appendix B also provides empirical results for predictive models such as i) logistic regression, ii) support vector machines and iii) neural networks. In response to your feedback, we have more prominently highlighted these results in the main paper by providing a short summary of results from these experiments in Section 6: Across all these models (i-iii), we observe that our suggested removal algorithms well outperform random guessing; often by up to 75 percentage points.

---

> > ### Author Response · Authors · 2022-11-12
> > **Follow up response to Reviewer 8JMg**
> >
> > > Hence, the novelty of this work is lowered as the explanation framework is another deletion-robust problem in disguise. Could you explain why is deletion robustness insufficient?
> >
> > As you have correctly understood, the robustness of algorithmic recourse depends on the robustness of the predictive model to data deletion requests. Therefore we totally agree that deletion robust models will lead to more robust recourse. In hindsight, i.e., after seeing our theoretical analysis, this may appear intuitive and clear. However, our work is the first to establish this fundamental link between data point influence and the robustness of counterfactual explanations. Moreover, it explicitly quantifies this link through theoretical results. This has not been demonstrated before and has real-world implications:
> >
> > - Our theory also suggests a fundamental paradigm shift in the way that the recourse literature thinks about robustness of algorithmic recourse: most of the literature follows the paradigm of developing robust recourse methods taking the predictive model as fixed [1-4]. This usually leads to recourses that are more difficult to act upon as they are less parsimonious and have higher recourse costs at the benefit of higher robustness [1-4]. In contrast to this paradigm, our analysis suggests that one can achieve an increased levels of robust recourse by training the model in such a way that has lower influence functions.
> >
> > - Thus, our theory can be operationalized with some approaches to minimize influence but which have been proposed in a different context, for instance private or deletion-robust models. Our work builds the bridge between these two fields, which we see as a valuable contribution.
> >
> > ----
> > References
> >
> > [1] E. Black, Z. Wang, M. Fredrikson, and A. Datta. Consistent counterfactuals for deep models. arXiv:2110.03109,, 2021
> >
> > [2] S. Upadhyay, S. Joshi, and H. Lakkaraju. Towards robust and reliable algorithmic recourse, NeurIPS , volume 34, 2021
> >
> > [3] M. Pawelczyk, T. Datta, J. van-den Heuvel, G. Kasneci, and H. Lakkaraju. Algorithmic recourse in the face of noisy human responses. arXiv preprint arXiv:2203.06768 , 2022
> >
> > [4] R. Dominguez-Olmedo, A.-H. Karimi, and B. Schölkopf. On the adversarial robustness of causal algorithmic recourse, ICML, 2022.

---

> > ### Comment · Reviewer_8JMg · 2022-11-18
> > **Question 4**
> >
> > This is regarding the Question 4:
> >
> > If I understand correctly, the authors suggest making the weights corresponding to the deleted data instance exactly zero, by subtracting after inversion. However, I do not think this is equivalent to the weights calculated with the entries removed in the NTK matrix prior the matrix inversion. Please clarify or show otherwise.
> >
> > Thanks!

---

> > > ### Author Response · Authors · 2022-11-18
> > > **Follow up response to Question 4**
> > >
> > > Dear reviewer, thank you for your comment. In equation 21 we use a well known result from the literature (for example, see Zhang and Zhang [1] and the references therein). We have now emphasized this more clearly. For the convenience of the reviewer, we now also provide a proof of the correctness of the weight update in equation 21. We refer to Appendix A.4 for the complete proof. There we demonstrate that the NTK with the updated weight of size $n$ is equivalent to a NTK based on a weight of length $n-1$ which was computed directly using the $n-1$ training points and the analytical solution. Note that we do not explicitly make the weight zero, but the derivations show that for the equivalence to hold, the weight corresponding to the deleted instance has to take a value of zero. This is in line with the clarification we made previously.
> > >
> > > [1] Rui Zhang and Shihua Zhang.Rethinking influence functions of neural networks in the overparameterized regime. In Proceedings of the AAAI Conference on Artificial Intelligence (AAAI), 2021

---

### Official Review · Reviewer_9bdZ · 2022-11-03

**Confidence:** 4
**Correctness:** 4
**Technical Novelty And Significance:** 4
**Empirical Novelty And Significance:** 3
**Recommendation:** 8

**Clarity, Quality, Novelty And Reproducibility:**

### Clarity

The paper is mostly easy to follow. However, its a bit too dense at places and important details are missed. Specifically:

1. Section 6: What is the median score? Why not consider the predicted class label?
2. Section 6: “positive leaning prediction (above median) to a negative one (below median)”. Does that mean that only negatively predicted inputs were considered? Would the results change if the other class was also included?
3. Eq 4: Would $\check{x}_{\omega}$ still be compute even if $\check{x}_1$ remains a valid recourse? This seems like a very crucial point that should be explicitly discussed.
4. Eq 4: How are categorical features handled in this definition (e.g., using Gower distance)? Are the features scaled?
5. Eq 3: I initially missed the fact that $\check{x}_1$ refers to the recourse with w_1. It might be worth clarifying this.
6. Eq 3: Shouldn’t \Delta(\omega) also be a function of x?
7. Were the methods (e.g., DICE) configured to generate more than one counterfactual or just a single one?

### Quality
For a first paper introducing a concept, the quality of theoretical analysis (though preliminary) and experimental results seems sufficient.

### Novelty
To the best of my knowledge, potential conflicts between right-to-be-forgotten and right-to-explanation have not been explored in the prior work.

### Reproducibility
The models are simple and the hyperparameters etc are reported. Explainers are used from the open-source CARLA library.


**Strength And Weaknesses:**

### Strengths

1. The potential conflict between right-to-be-forgotten and right-to-explanation sounds like a very relevant and timely area to explore. To the best of my knowledge, the paper is the first one to explore this question.
2. The paper is generally easy to follow. The metrics proposed here are simple and intuitive (though some details could be explained better -- see comments below). Same goes for the proposed strategies for identifying the data points that would lead to the most change.

### Weaknesses

1. At times, there is too much information and the takeaways are not easy to follow. E.g., how was it decided that only logreg and NTK should make it to the main paper? Why not add at least a summary of the experiments with related models?

2. Some of the details about the experimental setup could use more discussion, e.g., what is the intuition behind the 'median" strategy in Section 6? Please see detailed comments under the Clarity section.

**Summary Of The Paper:**

The paper looks at two aspects of ethics-aware ML: right-to-be-forgotten and right-to-explanation, and makes the points that the two could be in conflict with each other. Specifically, a change in the model parameters resulting from the right-to-be-forgotten requests could lead to invalidation of previously provided counterfactual explanations. The paper proposes metrics to measure change in explanations and also proposes algorithms for identifying data points whose removal would lead to the largest changes.

**Summary Of The Review:**

Overall, the problem proposed in the paper is quite important and timely. To the best of my knowledge, the paper is the first one to look into it. The algorithms and experiments are somewhat preliminary but sufficient for a first paper in the area.

---

> ### Author Response · Authors · 2022-11-11
> **Response to Reviewer 9bdZ**
>
> We thank the reviewer for their insightful feedback and are very grateful that the reviewer appreciates the novelty and significance of our work.
>
> > [...] how was it decided that only logreg and NTK should make it to the main paper? Why not add at least a summary of the experiments with related models?
>
> After obtaining results for each model / recourse combination, we had to select a subset of the empirical results to not overload the figures for the sake of readability. We chose linear regression and NTKs for the main paper as they are in line with our theory.
>
> As you have correctly pointed out, in appendix B we also provide empirical results for predictive models such as (i) logistic regression, (ii) support vector machines and (iii) neural networks. In response to your feedback, we have more prominently highlighted the empirical results from the appendix in Section 6 by providing a short summary of these results: Across all these models (i-iii), we observe that our suggested removal algorithms well outperform random guessing; often by up to 75 percentage points. Thanks for the suggestion and your constructive feedback.
>
> >What is the median score? Why not consider the predicted class label?
>
> We use two regression and classification data sets to test our methods on regression (Admission and Heloc) and classification tasks (Compas and Diabetes). For example, on the admission dataset the goal is to predict the first-year grade average of an individual in college. This dataset does not have discrete labels, which is why we adapted the following reasoning: A counterfactual is found when the predicted first-year average score switches from ‘below median’ to ‘above median’. Inspired by Spooner et al. [1], we then count an invalidation if, after the model update, the score of a counterfactual switches back to ‘below median’. For details on this procedure, please confer appendix B.
>
> > Does that mean that only negatively predicted inputs were considered? Would the results change if the other class was also included?
>
> Yes, your understanding is right. While we have not tested the approaches for the opposite case, we do not expect the results to be considerably different from our findings.
>
> > Would $\check{x}_{\omega}$ still be compute even if $\check{x}_1$ remains a valid recourse? This seems like a very crucial point that should be explicitly discussed.
>
> For an individual, the lowest cost recourse is essential as it can be most easily acted upon. Thus, in the recourse literature, the principal goal is to find the recourse associated with lowest costs. Therefore, we are interested in how the low cost recourse changes even if the outdated recourse would remain valid. Thank you for raising this point, we have included a discussion in Section 3 of the revised version of our draft.
>
> > How are cat. features handled in this definition (e.g., using Gower distance)? Are the features scaled?
>
> In our work, we consider four datasets. Admission and Heloc neither contain binary nor categorical features. The Diabetes dataset only contains binary features. We processed the Compas dataset by aggregating categories to only feature binary variables as in [2,3]. For example, the feature describing a person’s ethnicity would be encoded “white versus all other”. Moreover, the Gower distance would be a suitable alternative when dealing with categorical features.
>
> We applied a StandardScaler to normalize the features before processing, i.e., we subtracted the mean and divided by the standard deviation for the numerical continuous features.
>
>
> > I initially missed the fact that $\check{x}$ refers to the recourse with $w_1$. It might be worth clarifying this.
>
> In response to your feedback, we have stressed this point again in Definition 1. Please refer to the updated version of our manuscript.
>
> > Shouldn’t $\Delta(\omega)$ also be a function of $x$?
>
> Yes, this is correct. As our focus was on the dependency on $\omega$, we have not explicitly stated the dependency on $\mathbf{x}$. To avoid confusion, we decided to write $\Delta_x(\omega)$ in the updated manuscript.
>
> > Were the methods (e.g., DICE) configured to generate more than one counterfactual or just a single one?
>
> Thanks for your question. For DICE, for every input, we have generated two different counterfactual explanations. Then we randomly pick either the first or second counterfactual to be the counterfactual assigned to the given input. We have provided details for this procedure in Appendix C.
>
>
>
> -----
>
> [1] T. Spooner, D. Dervovic, J. Long, J. Shepard, J. Chen, and D. Magazzeni. Counterfactual explanations for arbitrary regression models. arXiv preprint arXiv:2106.15212, 2021
>
> [2] S. Upadhyay, S. Joshi, and H. Lakkaraju. Towards robust and reliable algorithmic recourse. NeurIPS, 2021
>
> [3] M. Pawelczyk, S. Bielawski, J. Van den Heuvel, T. Richter, and G. Kasneci. Carla: A python library to benchmark algorithmic recourse and counterfactual explanation algorithms. NeurIPS, 2021

---

### Decision · Program_Chairs · 2023-01-20

**Decision:**

Accept: poster

**Justification For Why Not Higher Score:**

The theory is not the most surprising (classical concepts such as leverage play a role) and the resulting algorithm does not have guarantees per se as it leads to a combinatorial optimization problem without a tractable solution in general.

**Justification For Why Not Lower Score:**

It's an original take on the challenges to algorithmic recourse that is framed based on very plausible real-world considerations about (lack of) dataset persistence. It is presented in a very well-organised and detailed fashion.

**Metareview: Summary, Strengths And Weaknesses:**

The paper provides a theory on how algorithmic recourse may be affected by data deletions (framed as "right to be forgotten" requests) that change the predictive model over which recourse operates.

Strengths: an original take on the challenges to algorithmic recourse that is framed based on very plausible real-world considerations about (lack of) dataset persistence. It is presented in a very well-organised and detailed fashion.

Weaknesses: the theory is not the most surprising (classical concepts such as leverage play a role) and the resulting algorithm does not have guarantees per se as it leads to a combinatorial optimization problem without a tractable solution in general.

**Note From Pc:**

if the above contains the word "oral" or "spotlight" please see: "oral" presentation means -> notable-top-5% and "spotlight" means -> notable-top-25%. As stated in our emails, we are disassociating presentation type from AC recommendations